# Learning with AMIGo:
# Adversarially Motivated Intrinsic Goals

**Andres Campero***
Brain and Cognitive Sciences, MIT
Cambridge, USA
`campero@mit.edu`

**Roberta Raileanu**
New York University
New York, USA
`raileanu@cs.nyu.edu`

**Heinrich Küttler**
Facebook AI Research
London, UK
`hnr@fb.com`

**Joshua B. Tenenbaum**
Brain and Cognitive Sciences, MIT
Cambridge, USA
`jbt@mit.edu`

**Tim Rocktäschel**
University College London
& Facebook AI Research
London, UK
`rockt@fb.com`

**Edward Grefenstette**
University College London
& Facebook AI Research
London, UK
`egrefen@fb.com`

## Abstract

A key challenge for reinforcement learning (RL) consists of learning in environments with sparse extrinsic rewards. In contrast to current RL methods, humans are able to learn new skills with little or no reward by using various forms of intrinsic motivation. We propose **AMIGo**, a novel agent incorporating—as form of meta-learning—a goal-generating teacher that proposes **A**dversarially **M**otivated **I**ntrinsic **GO**als to train a goal-conditioned "student" policy in the absence of (or alongside) environment reward. Specifically, through a simple but effective "constructively adversarial" objective, the teacher learns to propose increasingly challenging—yet achievable—goals that allow the student to learn general skills for acting in a new environment, independent of the task to be solved. We show that our method generates a natural curriculum of self-proposed goals which ultimately allows the agent to solve challenging procedurally-generated tasks where other forms of intrinsic motivation and state-of-the-art RL methods fail.

## 1 Introduction

The success of Deep Reinforcement Learning (RL) on a wide range of tasks, while impressive, has so far been mostly confined to scenarios with reasonably dense rewards (e.g. Mnih et al., 2016; Vinyals et al., 2019), or to those where a perfect model of the environment can be used for search, such as the game of Go and others (e.g. Silver et al., 2016; Duan et al., 2016; Moravčík et al., 2017). Many real-world environments offer extremely sparse rewards, if any at all. In such environments, random exploration, which underpins many current RL approaches, is likely to not yield sufficient reward signal to train an agent, or be very sample inefficient as it requires the agent to stumble onto novel rewarding states by chance. In contrast, humans are capable of dealing with rewards that are sparse and lie far in the future. For example, to a child, the future adult life involving education, work, or marriage provides no useful reinforcement signal. Instead, children devote much of their time to play, generating objectives and posing challenges to themselves as a form of intrinsic motivation. Solving such self-proposed tasks encourages them to explore, experiment, and invent; sometimes, as in many games and fantasies, without any direct link to reality or to any source of extrinsic reward. This kind of intrinsic motivation might be a crucial feature to enable learning in real-world environments (Schulz, 2012) .

To address this discrepancy between naïve deep RL exploration strategies and human capabilities, we present a novel meta-learning method wherein part of the agent learns to self-propose **A**dversarially **M**otivated **I**ntrinsic **Go**als (AMIGo). In AMIGo, the agent is decomposed into a goal-generating teacher and a goal-conditioned student policy. The teacher acts as a constructive adversary to the

---

*Work done during an internship at Facebook AI Research.

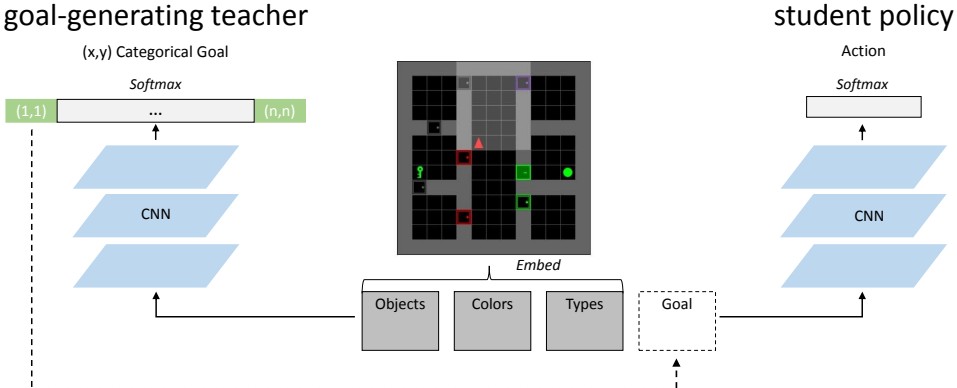

Figure 1: Training with AMIGO consists of combining two modules: a goal-generating teacher and a goal-conditioned student policy, whereby the teacher provides intrinsic goals to supplement the extrinsic goals from the environment. In our experimental set-up, the teacher is a dimensionality-preserving convolutional network which, at the beginning of an episode, outputs a location in absolute $(x, y)$ coordinates. These are provided as a one-hot indicator in an extra channel of the student's convolutional neural network, which in turn outputs the agent's actions.

student: the teacher is incentivized to propose goals that are not too easy for the student to achieve, but not impossible either. This results in a natural curriculum of increasingly harder intrinsic goals that challenge the agent and encourage learning about the dynamics of a given environment.

AMIGO can be viewed as an *augmentation* of any agent trained with policy gradient-based methods. Under this view, the original policy network becomes the student policy, which only requires its input-processing component to be adapted to accept an additional goal specification modality. The teacher policy can then be seen as a "bolt-on" to the original policy network, entailing that this method is—to the extent that the aforementioned goal-conditioning augmentation is possible—architecture-agnostic, and can be used on a variety of RL training model architectures and training settings.

As advocated in recent work (Cobbe et al., 2019; Zhong et al., 2020; Risi & Togelius, 2019; Küttler et al., 2020), we evaluate AMIGO for procedurally-generated environments instead of trying to learn to perform a specific task. Procedurally-generated environments are challenging since agents have to deal with a parameterized family of tasks, resulting in large observation spaces where memorizing trajectories is infeasible. Instead, agents have to learn policies that generalize across different environment layouts and transition dynamics (Rajeswaran et al., 2017; Machado et al., 2018; Foley et al., 2018; Zhang et al., 2018). Concretely, we use MiniGrid (Chevalier-Boisvert et al., 2018), a suite of fast-to-run procedurally-generated environments with a symbolic/discrete (expressed in terms of objects like walls, doors, keys, chests and balls) observation space which isolates the problem of exploration from that of visual perception. MiniGrid is a widely recognized challenging benchmark for intrinsic motivation, which was used in many recent publications such as Goyal *et al.* 2019, Bougie *et al.* 2019, Raileanu and Rocktäschel 2020, Modhe *et al.* 2020 etc. We evaluate our method on six different tasks from the MiniGrid domain with varying degrees of difficulties, in which the agent needs to acquire a diverse range of skills in order to succeed. Furthermore, MiniGrid is complex and competitive baselines such as IMPALA (that achieve SOTA in other domains like Atari) fail. Raileanu & Rocktäschel (2020) found that MiniGrid presents a particular challenge for existing state-of-the-art intrinsic motivation approaches. Here, AMIGO sets a new state-of-the-art on some of the hardest MiniGrid environments, being the only method capable of successfully obtaining extrinsic reward on some of them.

In summary, we make the following contributions: (i) we propose Adversarially Motivated Intrinsic GOals—an approach for learning a teacher that generates increasingly harder goals, (ii) we show, through 114 experiments on 6 challenging exploration tasks in procedurally generated environments, that agents trained with AMIGO gradually learn to interact with the environment and solve tasks which are too difficult for state-of-the-art methods, and (iii) we perform an extensive qualitative analysis and ablation study.

## 2 Related Work

Our work has connections to many different research areas but due to space constraints, we will focus our discussion on the most closely related topics, namely intrinsic motivation and curriculum learning.

**Intrinsic motivation** (Oudeyer et al., 2007; Oudeyer & Kaplan, 2009; Schmidhuber, 1991; Barto, 2013) methods have proven effective for solving various hard-exploration tasks (Bellemare et al., 2016; Pathak et al., 2017; Burda et al., 2019b). One prominent formulation is the use of *novelty*, which in its simplest form can be estimated with state visitation counts (Strehl & Littman, 2008) and has been extended to high-dimensional state spaces (Bellemare et al., 2016; Burda et al., 2019b; Ostrovski et al., 2017). Other sophisticated versions of *curiosity* (Schmidhuber, 1991) guide the agent to learn about environment dynamics by encouraging it to take actions that reduce the agent's uncertainty (Stadie et al., 2015; Burda et al., 2019b), have unpredictable consequences (Pathak et al., 2017; Burda et al., 2019a), or a large impact on the environment (Raileanu & Rocktäschel, 2020). Other forms of intrinsic motivation include *empowerment* (Klyubin et al., 2005) which encourages control of the environment by the agent, and *goal diversity* (Pong et al., 2019) which encourages maximizing the entropy of the goal distribution. In Lair et al. (2019), intrinsic goals are discovered from language supervision. The optimal rewards framework presents intrinsic motivation as a mechanism that goes beyond exploration, placing its origin in an evolutionary context (Singh et al., 2009), or framing it as a meta-optimization problem of selecting internal agent goals which optimize the designer's goals Sorg et al. (2010). More recently Zheng et al. (2018) extend this framework to learn parametric additive intrinsic rewards. Our work differs from all of the above by formulating intrinsic motivation as a "constructively adversarial" teacher that proposes increasingly harder goals for the agent.

**Curriculum learning** (Bengio et al., 2009) is another useful technique for tackling complex tasks but the curricula are typically handcrafted which can be time consuming. In our work, the curriculum is generated automatically in an unsupervised fashion. Another automatic curriculum approach learning was proposed by Schmidhuber (2011), where an agent constantly searches the space of problems for the next solvable one. However, this method is not scalable to more complex tasks. Florensa et al. (2017) generate a curriculum by increasing the distance of the starting-point to a goal. In contrast to AMIGO, this method assumes knowledge of the goal location and the ability to reset the agent in any state. A student can also self-propose a goal by hindsight experience replay (HER, Andrychowicz et al., 2017), which has been demonstrated to be effective in alleviating the sparse reward problem. Recent extensions have improved goal selection by balancing the difficulty and diversity (Fang et al., 2019) of goals. In contrast to our work, in HER, there is no explicit incentive for the agent to explore beyond its current reach. Since HER is rewarded for all the states it visits, it is rewarded for easy-to-reach states, even late in the training process. Other related work has trained a teacher to generate a curriculum of environments that maximize the learning process of the student (Portelas et al., 2020a). The question of the complementarity between these and our work is worth pursuing in the future. More similar to our work, Matiisen et al. (2017) train a teacher to select tasks in which the student is improving the most or in which the student's performance is decreasing to avoid forgetting. Note that AMIGO uses a different objective for training the teacher, which encourages the agent to solve progressively harder tasks. Similarly, Racaniere et al. (2019) train a goal-conditioned policy and a goal-setter network in a non-adversarial way to propose feasible, valid and diverse goals. Their feasibility criteria is similar to ours, but requires training an additional discriminator to rank the difficulty of the goals, while our teacher is directly trained to generate goals with an appropriate level of difficulty.[1] Recent surveys of curriculum generation in the context of RL include Narvekar et al. (2020) and Portelas et al. (2020b).

Closer to our work, Sukhbaatar et al. (2017) use an adversarial framework but require two modules that independently act and learn in the environment, where one module is encouraged to propose challenges to the other. This setup can be costly and is restricted to only proposing goals which have already been reached by the policy. Moreover, it requires a resettable or reversible environment. In

---

[1]Unfortunately, both the code for their method—which is far from simple—and for their experimental settings have not been made available by the authors. Therefore we not only cannot run a fair implementation of their approach against our setting for comparison, we cannot be guaranteed to successfully reimplement it ourselves as there is no way of reproducing their results in their setting without the code for the latter.

contrast, our method uses a single agent acting in the environment, and the teacher is less constrained in the space of goals it can propose.

Also similar to ours, Florensa et al. (2018) present GoalGAN, a generator that proposes goals with the appropriate level of difficulty as determined by a learned discriminator. While their work is similar in spirit with ours, there are several key differences. First, GoalGAN was created for and tested on locomotion tasks with continuous goals, whereas our method is designed for discrete action and goal spaces. While not impossible, adapting it to our setting is not trivial due to the GAN objective. Second, the authors do not condition the generator on the observation which is necessary in procedurally-generated environments that change with each episode. GoalGAN generates goals from a buffer, but previous goals can be unfeasible or nonsensical for the current episode. Hence, GoalGAN cannot be easily adapted to procedurally-generated environments.

A concurrent effort, Zhang et al. (2020) complements ours, but in the context of continuous control, by also generating a curriculum of goals which are neither too hard nor too easy using a measure of epistemic uncertainty based on an ensemble of value functions. This requires training multiple networks, which can become too computationally expensive for certain applications.

Finally, our approach is loosely inspired by generative adversarial networks (GANs, Goodfellow et al., 2014), where a generative model is trained to fool a discriminator which is trained to differentiate between the generated and the original examples. In contrast with GANs, AMIGo does not require a discriminator, and is "constructively adversarial", in that the goal-generating teacher is incentivized by its objective to propose goals which are challenging yet feasible for the student.

## 3 ADVERSARIALLY MOTIVATED INTRINSIC GOALS

AMIGo is composed of two subsystems: a goal-conditioned student policy which "controls" the agent's actions in the environment, and a goal-generating teacher (see Figure 1) which guides the student's training. The teacher proposes goals and is rewarded only when the student reaches the goal after a certain number of steps. The student receives reward for reaching the goal proposed by the teacher (discounted by the number of steps needed to reach the goal). The two components are trained adversarially in that the student maximizes reward by reaching goals as fast as possible, while the teacher maximizes reward by proposing goals which the student can reach, though not too quickly. In addition to this intrinsic reward, both modules are rewarded when the agent solves the full task.

### 3.1 TRAINING THE STUDENT

We consider the traditional RL framework of a Markov Decision Process with a state space $S$, a set of actions $A$ and a transition function $p(s_{t+1}|s_t, a_t)$ which specifies the distribution over next states given a current state and action. At each time-step $t$, the agent in state $s_t \in S$ takes an action $a_t \in A$ by sampling from a goal-conditioned stochastic student policy $\pi(a_t|s_t, g; \theta_\pi)$ where $g$ is a intrinsic goal provided by the teacher.

The teacher $G(s_0; \theta_g)$ is a separate policy, operating on a different "granularity" than the student: it takes as input an initial state and outputs as actions a goal $g$ for the student, which stays the same until a new goal is proposed. The teacher proposes a new goal every time an episode begins or whenever the student reaches the intrinsic goal. We assume that some goal verification function $v(s, g)$ can be specified as an indicator over whether a goal $g$ is achieved in a state $s$. We use this to define the undiscounted intrinsic reward $r_t^g$ as:

$$r_t^g = v(s_t, g) = \begin{cases} +1 & \text{if the state } s_t \text{ satisfies the goal } g \\ 0 & \text{otherwise} \end{cases}$$

At each time step $t$, the student receives a reward $r_t = r_t^g + r_t^e$, which is the sum of the intrinsic reward $r_t^g$ provided by the teacher and the extrinsic reward $r_t^e$ provided by the environment. The student, represented as a neural network with parameters $\theta_\pi$, is trained to maximize the discounted expected reward $R_t = \mathbb{E}\left[\sum_{k=0}^{H} \gamma^k r_{t+k}\right]$ where $\gamma \in [0, 1)$ is the discount factor. We consider a finite time horizon $H$ as provided by the environment.

## 3.2 Training the Teacher

The teacher $G(s_0; \theta_g)$, represented as a neural network with parameters $\theta_g$, is trained to maximise its expected reward. The teacher's reward $r^T$ is a function of the student's performance on the proposed goal and is computed every time this goal is reached (or at the end of an episode). As a result, the teacher operates at a different temporal frequency, and thus its rewards are not discounted according to the number of steps taken by the agent. To generate an automatic curriculum for the student, we positively reward the teacher if the student achieves the goal with suitable effort, but penalize it if the student either cannot achieve the goal, or can do so too easily. There are different options for measuring the performance of the student here, but for simplicity we will use the number of steps $t^+$ it takes the student to reach an intrinsic goal since the intrinsic goal was set (with $t^+ = 0$ if the student does not reach the goal before the episode ends). We define a threshold $t^*$ such that the teacher is positively rewarded by $r^T$ when the student takes more steps than the threshold to reach the set goal, and negatively if it takes fewer steps or never reaches the goal before the episode ends. We thus define the teacher reward as follows, where $\alpha$ and $\beta$ are hyperparameters (see Section 4.2 for implementation details) specifying the weight of positive and negative teacher reward:

$$r^T = \begin{cases} +\alpha & \text{if} \quad t^+ \geq t^* \\ -\beta & \text{if} \quad t^+ < t^* \end{cases}$$

One can try to calibrate a fixed target threshold $t^*$ to force the teacher to propose increasingly more challenging goals as the student improves. Initial experiments with a fixed threshold indicated that the loss function was sufficient to induce harder goals and curriculum learning. However, this threshold is different across environments and has to be carefully fixed (depending on the size and complexity of the environment). A more adaptive—albeit heuristic—approach we adopt is to linearly increase the threshold $t^*$ after a fixed number of times in which the student successfully reaches the intrinsic goals. Specifically, the threshold $t^*$ is increased by 1 whenever the student successfully reaches an intrinsic goal in more than $t^*$ steps for ten times in a row. This increase in the target threshold provides an additional metric to visualize the improvement of the student through the "difficulty" of its goals (see Figure 3).

## 3.3 Types of Goals

We can conceive of variants of AMIGo whereby goals are provided in the form of linguistic instructions, images, etc. To prove the concept, in this framework, a goal is formally defined as a change in the observation on a tile, as specified by an $(x, y)$ coordinate. The agent must modify the tile before the end of an episode (e.g. by moving to it, or causing the object in it to move or change state). The verification function $v$ is then trivially the indicator function of whether the cell state is different from its initial state at the beginning of the episode. Proposing $(x, y)$ coordinates as goals can present a diverse set of ways for an agent to achieve the goal, as the coordinates can not only be affected by reaching them but also by modifying what is on them. This includes picking up keys, opening doors, and dropping objects onto empty tiles. In some cases in our setting, moving over a square is not the simplest thing possible (e.g. when an obstacle can be removed or a door can be opened). Similarly, in other tasks and environments it could be easier to affect a cell by throwing something at it, rather than by reaching it. Likewise, simply navigating to a set of coordinates (say, the corner of a locked room) might require solving several non-trivial sub-problems (e.g. identifying the right key, going to it, then going to the door, unlocking it, and finally going to the target location). We give some examples of goals proposed by the teacher, alongside the progression in their difficulty as the student improves, in Figure 3.

## 3.4 Auxiliary Teacher Losses

To complement our main form of intrinsic reward, we explore a few other criteria, including goal diversity, extrinsic reward, environment change and novelty. We report, in our experiments of Section 4, the results for AMIGo using these auxiliary losses. We present, in Appendix D, an ablation study of the effect of these losses, alongside some alternatives to the reward structure for the teacher network.

**Diverse Goals.** One desirable property is goal diversity (Pong et al., 2019; Raileanu & Rocktäschel, 2020). In our implementation of AMIGo in the experiments of Section 4, we used entropy regulariza-

tion to train the teacher and student, which encourages such diversity. This regularization, along with the scheduling of the threshold, helps the teacher avoid getting stuck in local minima. Additionally, we considered rewarding the teacher for proposing novel goals similar to count-based exploration methods (Bellemare et al., 2016; Ostrovski et al., 2017) with the difference that in our case the counts are for goals instead of states, based on the number of times the teacher presents a type of goal to the student. This did not improve performance and is not part of our model for the rest of the paper.

**Episode Boundary Awareness.** When playing in a procedurally-generated environment, humans will notice the factors of variation and exploit them. In episodic training, RL agents and algorithms are informed if a particular state was an episode end. To bias AMIGo towards learning the factors of variation in an environment, while not giving it any domain knowledge or any privileged information which other comparable intrinsic motivation systems and RL agents would not have access to, we positively reward the teacher if the content of the goal location it proposes changes at an episode boundary, regardless of whether this change was due to the agent. Thus, the teacher is rewarded for selecting goals where the object type changes if the episode changes (for example a door becomes a wall, or a key becomes an empty tile due to the new episode configuration). While this heuristic is quite general and could be effective for many tasks as it encourages agents to note environmental factors of variation, we note it might not be useful in all possible domains and as such is not an essential part of AMIGo. A comparison an extension of this loss other intrinsic motivation methods would interesting, but is not straightforward and is left for future research, we just note that this auxiliary loss on its own is not able to solve even the medium difficulty environments.

**Extrinsic Goals.** To help transition into the extrinsic task and avoid local minima, we reward both the teacher and the student with environment reward whenever the student reaches the extrinsic goal, even if this does not coincide with the intrinsic goal set by the teacher. This avoids the degenerate case where the student becomes good at satisfying the extrinsic goal, and the teacher is forced to encourage it "away" from it.

## 4 EXPERIMENTS

We follow Raileanu & Rocktäschel (2020) and evaluate our models on several challenging procedurally-generated environments from MiniGrid (Chevalier-Boisvert et al., 2018). This environment provides a good testbed for exploration in RL since the observations are symbolic rather than high-dimensional, which helps to disentangle the problem of exploration from that of visual understanding. We compare AMIGo with state-of-the-art methods that use various forms of exploration bonuses. We use TorchBeast (Küttler et al., 2019), a PyTorch platform for RL research based on IMPALA (Espeholt et al., 2018) for fast, asynchronous parallel training. The code for these experiments is included in the supplementary materials, and has also been released under "https://anonymous" to facilitate reproduction of our method and its use in other projects.

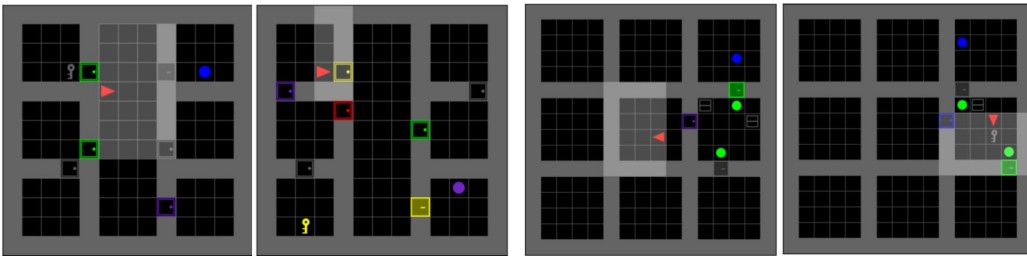

(a) Example episodes in **KCharder**.  (b) Example episodes in **OMhard**.

Figure 2: Examples of MiniGrid environments. **KCharder** requires finding the key that can unlock a door which blocks the room where the goal is (the blue ball). **OMhard** requires a sequence of correct steps usually involving opening a door, opening a chest to find a key of the correct color, picking-up the key to open the door, and opening the door to reach the goal. The configuration and colors of the objects change from one episode to another. To our knowledge, AMIGo is the only algorithm that can solve these tasks. For other examples, see the MiniGrid repository.

## 4.1 ENVIRONMENTS

We evaluate AMIGO on the following MiniGrid environments: KeyCorrS3R3 (**KCmedium**), ObstrMaze1Dl (**OMmedium**), ObstrMaze2Dlhb (**OMmedhard**), KeyCorrS4R3 (**KChard**), Key-CorrS5R3 (**KCharder**), and ObstrMaze1Q (**OMhard**). The agent receives a full observation of the MiniGrid environment. The layout of the environment changes at every episode as it is procedurally-generated. Examples of these tasks can be found in Figure 2. Each environment is a grid of size $N \times N$ ($N$ being environment-specific) where each tile contains at most one of the following colored objects: wall, door, key, ball, chest. An object in each episode is selected as an extrinsic goal. If the agent reaches the extrinsic goal, or a maximum number of time-steps is reached, the environment is reset. The agent can take the following actions: turn left, turn right, move forward, pick up an object, drop an object, or toggle (open doors or interact with objects). Each tile is encoded using three integer values: the object, the color, and a type or flag indicating whether doors are open or closed. While policies could be learned from pixel observation alone, we will see below that the exploration problem is sufficiently complex with these semantic layers, owing to the procedurally generated nature of the tasks. The observations are transformed before being fed to agents by embedding each tile of the observed frame into a single representation encoding the object type, color, and type/flag.

The extrinsic reward provided by each environment for reaching the extrinsic goal in $t$ steps is $r_t^e = 1 - (.9 \cdot t)/t^{\max}$, where $t^{\max}$ is the maximum episode length (which is intrinsic to each environment and set by the MiniGrid designers), if the extrinsic goal is reached at $t$, and 0 otherwise. Episodes end when the goal is reached, and thus the scale of the positive reward encourages agents to reach the goal as quickly as possible.

## 4.2 AMIGO IMPLEMENTATION

The teacher is a dimensionality-preserving network of four convolutional layers interleaved with exponential linear units. Similarly, the student consists of four convolutional layers interleaved with exponential linear units followed by two linear layers with rectified linear units. Both the student and the teacher are trained using the TorchBeast (Küttler et al., 2019) implementation of IMPALA (Espeholt et al., 2018), a distributed actor-critic algorithm. But while the teacher proposes goals only at the beginning of an episode or when the student reaches a goal, the student produces an action and gets a reward at every step. To replicate the structure of reward for reaching extrinsic goals, intrinsic reward for the student is discounted to $r_t^g = 1 - (.9 \cdot t)/t^{\max}$ when $v(s_t, g) = 1$, and 0 otherwise. The hyperparameters for the reward for the teacher $r^T$ are grid searched, and optimal values are found at $\alpha = .7$ and $\beta = .3$ (see Appendix B for full hyperparameter search details).

## 4.3 BASELINES AND EVALUATION

We use **IMPALA** (Espeholt et al., 2018) without intrinsic motivation as a standard deep RL baseline. We then compare AMIGO to a series of methods that use intrinsic motivation to supplement extrinsic reward, as listed here. **Count** is Count-Based Exploration from Bellemare et al. (2016), which computes state visitation counts and gives higher rewards to less visited states. **RND** is Random Network Distillation Exploration by Burda et al. (2019b) which uses a random network to compute a prediction error used as a bonus to reward novel states; **ICM** is Intrinsic Curiosity Module from Pathak et al. (2017), which trains forward and inverse models to learn a latent representation used to compare the predicted and actual next states. The Euclidean distance between the representations of predicted and actual states (as measured in the latent space) is used as intrinsic reward. **RIDE**, from Raileanu & Rocktäschel (2020), defines the intrinsic reward as the (magnitude of the) change between two consecutive state representations.

We have noted from the literature that some of these baselines were designed for partially observable environments (Raileanu & Rocktäschel, 2020; Pathak et al., 2017) so they might benefit from observing an agent-centric partial view of the environment rather than a full absolute view (Ye et al., 2020). Despite our environment being fully observable, for the strongest comparison with AMIGO we ran the baselines in each of the following four modes: full observation of the environment for both the intrinsic reward module and the policy network, full observation for the intrinsic reward and partial observation for the policy, partial view for the intrinsic reward and full view for the policy, and partial view for both. We use an LSTM for the student policy network when it is provided with partial observations and a feed-forward network when provided with full observations. In

Section 4.4, we report the best result (across all four modes) for each baseline and environment pair, with a full breakdown of the results in Appendix A. This, alongside a comprehensive hyperparameter search, ensures that AMIGO is compared against the baselines trained under their individually best-performing training arrangement. We also compare AMIGO to the authors' implementation[2] of Asymmetric Self-Play (**ASP**) (Sukhbaatar et al., 2017). In their reversible mode two policies are trained adversarially: Alice starts from a start-point and tries to reach goals, while Bob is tasked to travel in reverse from the goal to the start-point.

We ran each experiment with five different seeds, and report in Section 4.4 the means and standard deviations. The full hyperparameter sweep for AMIGO and all baselines is reported in Appendix B, alongside best hyperparameters across experiments.

## 4.4 RESULTS AND DISCUSSION

We summarize the main results of our experiments in Table 1. As discussed in Section 4.3, the reported result for each baseline and each environment is that of the best performing configuration for the policy and intrinsic motivation system for that environment, as reported in Tables 2–5 of Appendix A. This aggregation of 114 experiments (not counting the number of times experiments were run for different seeds) ensures that each baseline is given the opportunity to perform in its best setting, in order to fairly benchmark the performance of AMIGO.

Table 1: Comparison of Mean Extrinsic Reward at the end of training (averaging over a batch of episodes as in IMPALA). Each entry shows the result of the best observation configuration, for each baseline, from Tables 2–5 of Appendix A.

| Model | Medium Difficulty Environments | | | Hard Environments | | |
|---|---|---|---|---|---|---|
| | KCmedium | OMmedium | OMmedhard | KChard | KCharder | OMhard |
| AMIGO | $\mathbf{.93} \pm .00$ | $.92 \pm .00$ | $.83 \pm .05$ | $\mathbf{.54} \pm .45$ | $\mathbf{.44} \pm .44$ | $\mathbf{.17} \pm .34$ |
| IMPALA | $.00 \pm .00$ | $.00 \pm .00$ | $.00 \pm .00$ | $.00 \pm .00$ | $.00 \pm .00$ | $.00 \pm .00$ |
| RND | $.89 \pm .00$ | $\mathbf{.94} \pm .00$ | $\mathbf{.88} \pm .03$ | $.23 \pm .40$ | $.00 \pm .00$ | $.00 \pm .00$ |
| RIDE | $.90 \pm .00$ | $\mathbf{.94} \pm .00$ | $.86 \pm .06$ | $.19 \pm .37$ | $.00 \pm .00$ | $.00 \pm .00$ |
| COUNT | $.90 \pm .00$ | $.04 \pm .04$ | $.00 \pm .00$ | $.00 \pm .00$ | $.00 \pm .00$ | $.00 \pm .00$ |
| ICM | $.42 \pm .21$ | $.19 \pm .19$ | $.16 \pm .32$ | $.00 \pm .00$ | $.00 \pm .00$ | $.00 \pm .00$ |
| ASP | $.00 \pm .00$ | $.00 \pm .00$ | $.00 \pm .00$ | $.00 \pm .00$ | $.00 \pm .00$ | $.00 \pm .00$ |

IMPALA and Asymmetric Self-Play are unable to pass any of these medium or hard environments. ICM and Count struggle on the "easier" medium environments, and fail to obtain any reward from the hard ones. Only RND and RIDE perform competitively on the medium environments, but struggle to obtain any reward on the harder environments.

Our results demonstrate that AMIGO establishes a new state of the art in harder exploration problems in MiniGrid. On environments with medium difficulty such as **KCmedium**, **OMmedium**, and **OMmedhard**, AMIGO performs comparably to other state-of-the-art intrinsic motivation methods. AMIGO is often able to successfully reach the extrinsic goal even on the hardest tasks. To showcase results and sample complexity, we illustrate and discuss how mean extrinsic reward changes during training in Appendix C. To analyze which components of the teacher loss were important, we present, in Appendix D, an ablation study over the components presented in Section 3.4. Qualitatively, the learning trajectories of AMIGO display interesting and partially adversarial dynamics. These often involve periods in which both modules cooperate as the student becomes able to reach the proposed goals, followed by others in which the student becomes too good, forcing a drop in the teacher reward, in turn forcing the teacher to increase the difficulty of the proposed goals and forcing the student to further explore. In Appendix E, we provide a more thorough qualitative analysis of AMIGO, wherein we describe the different phases of evolution in the difficulty of the intrinsic goals proposed by the teacher, as exemplified in Figure 3. Further goal examples are shown in Figure 6 of Appendix F.

---

[2]https://github.com/tesatory/hsp

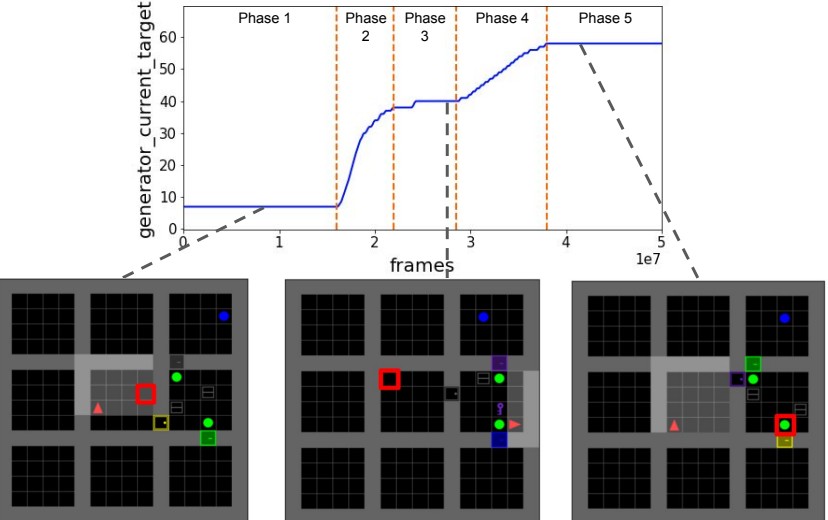

Figure 3: Examples of a curriculum of goals proposed for different episodes of a particular learning trajectory on **OMhard**. The red triangle is the agent, the red square is the goal proposed by the teacher, and the blue ball is the extrinsic goal. The top panel shows the threshold target difficulty, $t^*$ of the goals proposed by the teacher. The teacher first proposes very easy nearby goals, then it learns to propose goals that involve traversing rooms and opening doors, while in the third phase the teacher proposes goals which involve removing obstacles and interacting with objects.

## 5    CONCLUSION

In this work, we propose AMIGO, a meta-learning framework for generating a natural curriculum of goals that help train an agent as a form of intrinsic reward, to supplement extrinsic reward (or replace it if it is not available). This is achieved by having a goal generator as a teacher that acts as a constructive adversary, and a policy that acts as a student conditioning on those goals to maximize an intrinsic reward. The teacher is rewarded to propose goals that are challenging but not impossible. We demonstrate that AMIGO surpasses state-of-the-art intrinsic motivation methods in challenging procedurally-generated tasks in a comprehensive comparison against multiple competitive baselines, in a series of 114 experiments across 6 tasks. Crucially, it is the only intrinsic motivation method which allows agents to obtain any reward on some of the harder tasks, where non-intrinsic RL also fails.

The key contribution of this paper is a model-agnostic framework for improving the sample complexity and efficacy of RL algorithms in solving the exploration problems they face. In our experiments, the choice of goal type imposed certain constraints on the nature of the observation, in that both the teacher and student need to fully observe the environment, due to the goals being provided as absolute coordinates. Technically, this method could also be applied to partially observed environments where part of the full observation is uncertain or occluded (e.g. "fog of war" in StarCraft), as the only requirement is that absolute coordinates can be provided and acted on. However, this is not a fundamental requirement, and in future work we would wish to investigate the cases where the teacher could provide more abstract goals, perhaps in the form of language instructions which could directly specify sequences of subgoals. Other extensions to this work worth investigating are its applicability to continuous control domains, visually rich domains, or more complex procedurally generated environments such as (Cobbe et al., 2019). Until then, we are confident we have proved the concept in a meaningful way, which other researchers will already be able to easily adapt to their model and RL algorithm of choice, in their domain of choice.

### ACKNOWLEDGMENTS

We thank the anonymous reviewers for their candid and helpful feedback and discussions. Roberta was supported by the DARPA L2M grant.

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

Table 2: Fully observed intrinsic reward, fully observed policy.

| Model | Medium Difficulty Environments | | | Hard Environments | | |
|-------|--------|----------|-----------|--------|----------|--------|
|       | KCmedium | OMmedium | OMmedhard | KChard | KCharder | OMhard |
| AMIGo | **.93** ± .00 | **.92** ± .00 | **.83** ± .05 | **.54** ± .45 | **.44** ± .44 | **.17** ± .34 |
| IMPALA | .00 ± .00 | .00 ± .00 | .00 ± .00 | .00 ± .00 | .00 ± .00 | .00 ± .00 |
| RND | .00 ± .00 | .00 ± .00 | .00 ± .00 | .00 ± .00 | .00 ± .00 | .00 ± .00 |
| RIDE | .00 ± .00 | .04 ± .04 | .00 ± .00 | .00 ± .00 | .00 ± .00 | .00 ± .00 |
| COUNT | .00 ± .00 | .00 ± .00 | .00 ± .00 | .00 ± .00 | .00 ± .00 | .00 ± .00 |
| ICM | .00 ± .00 | .01 ± .01 | .00 ± .00 | .00 ± .00 | .00 ± .00 | .00 ± .00 |
| ASP | .00 ± .00 | .00 ± .00 | .00 ± .00 | .00 ± .00 | .00 ± .00 | .00 ± .00 |

Table 3: Partially observed intrinsic reward, fully observed policy.

| Model | Medium Difficulty Environments | | | Hard Environments | | |
|-------|--------|----------|-----------|--------|----------|--------|
|       | KCmedium | OMmedium | OMmedhard | KChard | KCharder | OMhard |
| RND | .64 ± .09 | **.01** ± .01 | .00 ± .00 | .00 ± .00 | .00 ± .00 | .00 ± .00 |
| RIDE | **.84** ± .02 | .00 ± .00 | .00 ± .00 | .00 ± .00 | .00 ± .00 | .00 ± .00 |
| COUNT | .45 ± .26 | .00 ± .00 | .00 ± .00 | .00 ± .00 | .00 ± .00 | .00 ± .00 |
| ICM | .42 ± .21 | .00 ± .00 | .00 ± .00 | .00 ± .00 | .00 ± .00 | .00 ± .00 |

Table 4: Fully observed intrinsic reward, partially observed policy.

| Model | Medium Difficulty Environments | | | Hard Environments | | |
|-------|--------|----------|-----------|--------|----------|--------|
|       | KCmedium | OMmedium | OMmedhard | KChard | KCharder | OMhard |
| RND | .00 ± .00 | .00 ± .00 | .00 ± .00 | .00 ± .00 | .00 ± .00 | .00 ± .00 |
| RIDE | **.88** ± .01 | **.94** ± .00 | **.18** ± .35 | **.19** ± .37 | .00 ± .00 | .00 ± .00 |
| COUNT | .01 ± .01 | .00 ± .00 | .00 ± .00 | .00 ± .00 | .00 ± .00 | .00 ± .00 |
| ICM | .06 ± .12 | .05 ± .06 | .16 ± .32 | .00 ± .00 | .00 ± .00 | .00 ± .00 |

Table 5: Partially observed intrinsic reward, partially observed policy.

| Model | Medium Difficulty Environments | | | Hard Environments | | |
|-------|--------|----------|-----------|--------|----------|--------|
|       | KCmedium | OMmedium | OMmedhard | KChard | KCharder | OMhard |
| RND | .89 ± .00 | **.94** ± .00 | **.88** ± .03 | **.23** ± .40 | .00 ± .00 | .00 ± .00 |
| RIDE | **.90** ± .00 | .85 ± .28 | .86 ± .06 | .00 ± .00 | .00 ± .00 | .00 ± .00 |
| COUNT | **.90** ± .00 | .04 ± .04 | .00 ± .00 | .00 ± .00 | .00 ± .00 | .00 ± .00 |
| ICM | .00 ± .00 | .19 ± .19 | .00 ± .00 | .00 ± .00 | .00 ± .00 | .00 ± .00 |

# A  FULL RESULTS

Tables 2–5 show the final performance of the intrinsic motivation baselines trained using one of four different training regimes enumerated in Section 4.3. For each baseline, we train on **KCMedium** and **OMmedium**, and use the best hyperparameters for each task (for that particular baseline and training regime) to train it on the remaining harder versions of those environments (i.e. on **KChard** and **KCharder** or **OMmedhard** and **OMhard**, respectively).

For IMPALA, the numbers reported for **KCmedium** and **OMmedium** are from the experiments in Raileanu & Rocktäschel (2020), while the numbers for the harder environments are presumed to be .00 because IMPALA fails to train on simpler environments.

As a sanity check, we also verified that ASP learns successfully in easier environments not considered here, such as MiniGrid-Empty-Random-5x5-v0, and MiniGrid-KeyCorridorS3R1-v0, to validate the official PyTorch implementation.

Tables 2–5 indicate that the best training regime for the intrinsic motivation baselines (for all the tasks they can reliably solve) is the one that uses a partially observed intrinsic reward and a partially observed policy (Table 5). When the intrinsic reward is based on a full view of the environment, Count and RND will consider almost all states to be "novel" since the environment is procedurally-generated. Thus, the reward they provide will not be very helpful for the agent since it does not transfer knowledge from one episode to another (as is the case in fixed environments (Bellemare et al., 2016; Burda et al., 2019b)). In the case of RIDE and ICM, the change in the full view of the environment produced by one action is typically a single number in the MiniGrid observation. For ICM, this means that the agent can easily learn to predict the next state representation, so the intrinsic reward might vanish early in training leaving the agent without any guidance for exploring (Raileanu & Rocktäschel, 2020). For RIDE, it means that the intrinsic reward will be largely uniform across all state-action pairs, thus not differentiating between more and less "interesting" states (which it can do when the intrinsic reward is based on partial observations (Raileanu & Rocktäschel, 2020)).

## B  HYPERPARAMETER SWEEPS AND BEST VALUES

For AMIGO, we grid search over batch size for student and teacher $\in \{8, 32, 150\}$, learning rate for student $\in \{.001, .0001\}$, learning rate for teacher $\in \{.05, .01, .0001\}$ unroll length $\in \{50, 100\}$, entropy cost for student $\in \{.0005, .001, .0001\}$, entropy cost for teacher $\in \{.001, .01, .05\}$, embedding dimensions for the observations $\in \{5, 10, 20\}$, embedding dimensions for the student last linear layer $\in \{128, 256\}$, and teacher loss function parameters $\alpha$ and $\beta \in \{1.0, 0.7, 0.5, 0.3, 0.0\}$.

For RND, RIDE, Count, and ICM, we used learning rate $10^{-4}$, batch size 32, unroll length 100, RMSProp optimizer with $\epsilon = 0.01$ and momentum 0, which were the best values found for these methods on MiniGrid tasks by Raileanu & Rocktäschel (2020). We further searched over the entropy coefficient $\in \{0.0005, 0.001, 0.0001\}$ and the intrinsic reward coefficient $\in \{0.1, 0.01, 0.5\}$ on KCmedium and OMmedium. The results reported in Tables 2, 3 and 4 use the best values found from these experiments, while the results reported in Table 5 use the best parameter values reported by Raileanu & Rocktäschel (2020). For ASP, we ran the authors' implementation using its reverse mode. We used the defaults for most hyperparameters, grid searching only over sp_steps $\in \{5, 10, 20\}$, sp_test_rate $\in \{.1, .5\}$, and sp_alice_entr $\in \{.003, .03\}$.

The best hyperparameters for AMIGO and each baseline are reported below:

**AMIGO:** a student batch size of 8, a teacher batch size of 150, a student learning rate of .001, a teacher learning rate of .001, an unroll length of 100, a student entropy cost of .0005, a teacher entropy cost of .01, and observation embedding dimension of 5, a student last layer embedding dimension of 256, and finally, $\alpha = 0.7$ and $\beta = 0.3$.

**RND:** partially observed intrinsic reward, partially observed policy, entropy cost of .0005, intrinsic reward coefficient of .1

**RIDE:** for **KCmedium**, partially observed intrinsic reward, partially observed policy, entropy cost of .0005, intrinsic reward coefficient of .1; for **OMmedium**: fully observed intrinsic reward, partially observed policy, entropy cost of .0005, intrinsic reward coefficient of .1

**COUNT:** partially observed intrinsic reward, partially observed policy, entropy cost of .0005, intrinsic reward coefficient of .1

**ICM:** for **KCmedium**, partially observed intrinsic reward, fully observed policy, entropy cost of .0005, intrinsic reward coefficient of .1; for **OMmedium**: partially observed intrinsic reward, partially observed policy, entropy cost of .0005, intrinsic reward coefficient of .1

**ASP:** Best performing hyperparameters (in the easier environments) were 10 sp_steps, a sp_test_rate of .5, and Alice entropy of .003. All other hyperparameters used the defaults in the codebase.

## C  SAMPLE EFFICIENCY

We show, in Figure 4, the mean extrinsic reward over time during training for the best configuration of the various methods. The first row consists of intermediately difficult environments in which different forms of intrinsic motivation perform similarly, the first two evironments require less than 30 million

steps while **OMmedhard** and the three more challenging environments of the bottom rows require in the order of hundreds of millions of frames. Any plot where a method's line is not visible indicates that the method is consistently failing to reach reward states throughout its training.

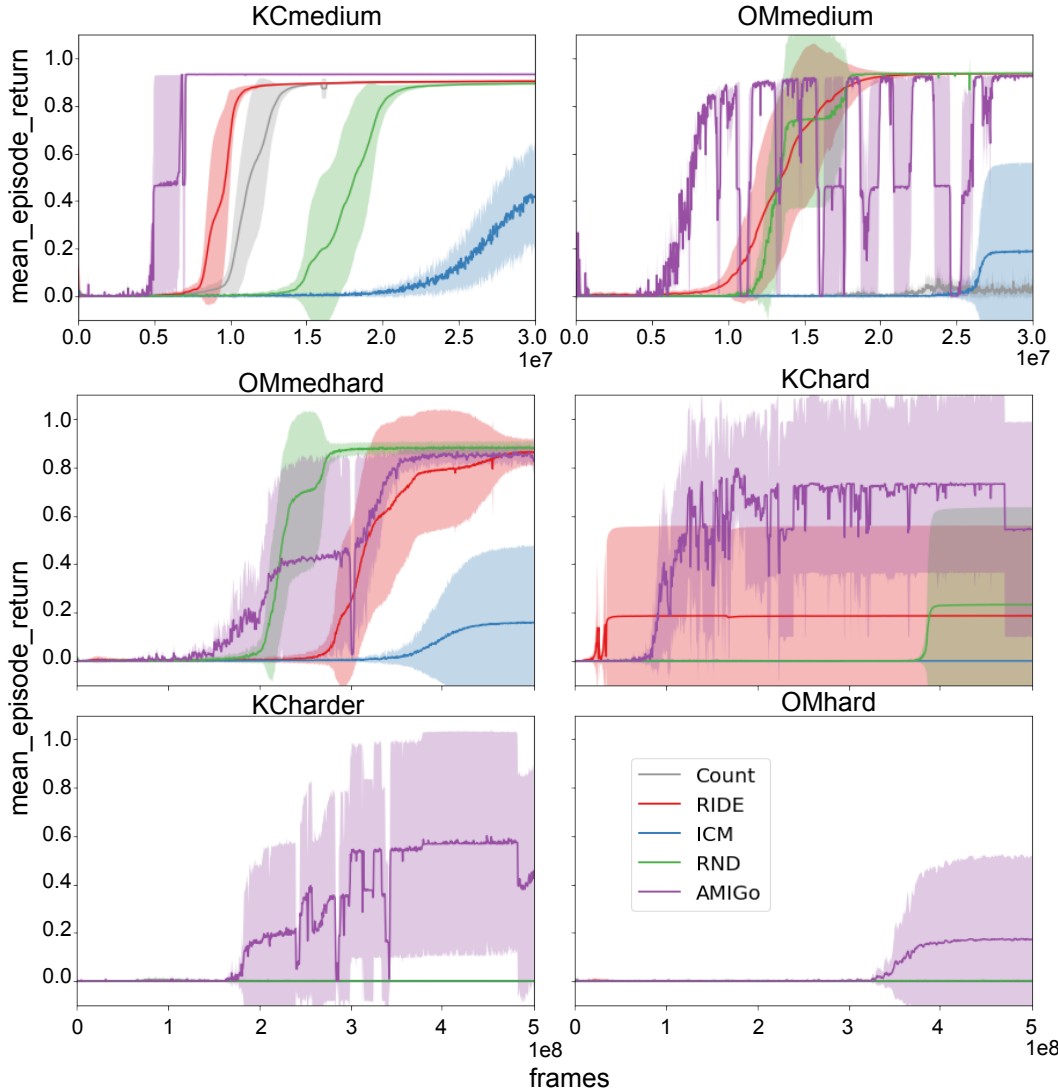

Figure 4: Reward curves over training time comparing AMIGO to competing methods and baselines. The y-axis shows the Mean Extrinsic Reward (performance) obtained in two medium and four harder different environments, shown for 30M and 500M frames respectively.

The important point of note here is that on the two easiest environments, **KCmedium** and **OMmedium**, agents need about 10 million steps to converge while on the other four more challenging environments, they need an order of 100 million steps to learn the tasks, showcasing AMIGO's contributions not just to solving the exploration problem, but also to improving the sample complexity of agent training.

# D    ABLATION STUDY

To further explore the effectiveness and robustness of our method, in this subsection we investigate the alternative criteria discussed in Section 3.4. We compare the FULL MODEL with its ablations and alternatives consisting of removing the extrinsic bonus (NOEXTRINSIC), removing the environment change bonus (NOENVCHANGE), adding a novelty bonus( WITHNOVELTY).

We also considered two alternative reward forms for the teacher to provide a more continuous and gradual reward than the previously introduced "all or nothing" threshold. We consider a **Gaussian** $p \sim \text{Normal}(t^*, \sigma)$ reward around the target threshold $t^*$:

$$r^T = \begin{cases} -1 & \text{if} \quad t^+ = 0 \\ 1 + \log_p(t^+) - \log_p(t^*) & \text{otherwise} \end{cases}$$

and a **Linear-Exponential** reward which grows linearly towards the threshold and then decays exponentially as the goal proposed becomes too hard (as measured according to the number of steps):

$$r^T = \begin{cases} e^{-(t^+ - t^*)/c} & \text{if} \quad t^+ \geq t^* \\ t^+/t^* & \text{if} \quad t^+ < t^* \end{cases}$$

We report these two alternative forms of reward as (GAUSSIAN and LINEAR-EXPONENTIAL) in the study below.

Table 6: Ablations and Alternatives. Number of steps (in millions) for models to learn to reach its final level of reward in the different environments (0 means the model did not learn to get any extrinsic reward). FULL MODEL is the main algorithm described above. NOEXTRINSIC does not provide any extrinsic reward to the teacher. NOENVCHANGE removes the reward for selecting goals that change as a result of episode resets. WITHNOVELTY adds a novelty bonus that decreases depending on the number of times an object has been successfully proposed. GAUSSIAN and LINEAR-EXPONENTIAL explore alternative reward functions for the teacher.

| Model | Medium Difficulty Environments | | | Hard Environments | | |
|---|---|---|---|---|---|---|
| | **KCmedium** | **OMmedium** | **OMmedhard** | **KChard** | **KCharder** | **OMhard** |
| FULL MODEL | **7M** | **8M** | **320M** | 140M | **300M** | **370M** |
| NOEXTRINSIC | 240M | 50M | 0 | 0 | 0 | 0 |
| NOENVCHANGE | 400M | 37M | 0 | 0 | 0 | 0 |
| WITHNOVELTY | 15M | 20M | 350M | **100M** | 370M | 0 |
| GAUSSIAN | 320M | 60M | 0 | 0 | 0 | 0 |
| LINEAR-EXP | 0 | 0 | 0 | 0 | 0 | 0 |

Performance is shown in Table 6 where the number of steps needed to converge (to the final extrinsic reward) is reported. A positive number means the model learned to solve the task, while 0 means the model did not manage to get any extrinsic reward. For all models we encourage goal diversity on the teacher with a high entropy coefficient of .05 (as compared to .0005 of the student).

As the table shows, removing the extrinsic reward or the environment change bonus severely hurts the model, making it unable to solve the harder environments. The novelty bonus was minimally beneficial in one of the environments (namely **KChard**) but slightly ineffective on the others. The more gradual reward forms considered are not robust to the learning dynamics and often result in the system going into rabbit holes where the algorithm learns to propose goals which provide sub-optimal rewards, thus not helping to solve the actual task. Best results across all environments in our Full Model were obtained using the simple threshold reward function along with entropy regularization and in combination with the extrinsic reward and changing bonuses, but without the novelty bonus.

## E    QUALITATIVE ANALYSIS

To better understand the learning dynamics of AMIGo, Figure 5 shows the intrinsic reward throughout training received by the student (top panel) as well as the teacher (middle panel). The bottom panel shows the difficulty of the proposed goals as measured by the target threshold $t^*$ used by the teacher (described in Section 3). The trajectories reflect interesting and complex learning dynamics.

For visualization purposes we divide this learning period into five phases: **Phase 1**: The student slowly becomes able to reach intrinsic goals with minimal difficulty. The teacher first learns to propose easy nearby goals. **Phase 2**: Once the student learns how to reach nearby goals, the adversarial dynamics cause a drop in the teacher reward which is then forced to explore and propose harder goals. **Phase 3**: An equilibrium is found in which the student is forced to learn to reach more challenging goals.

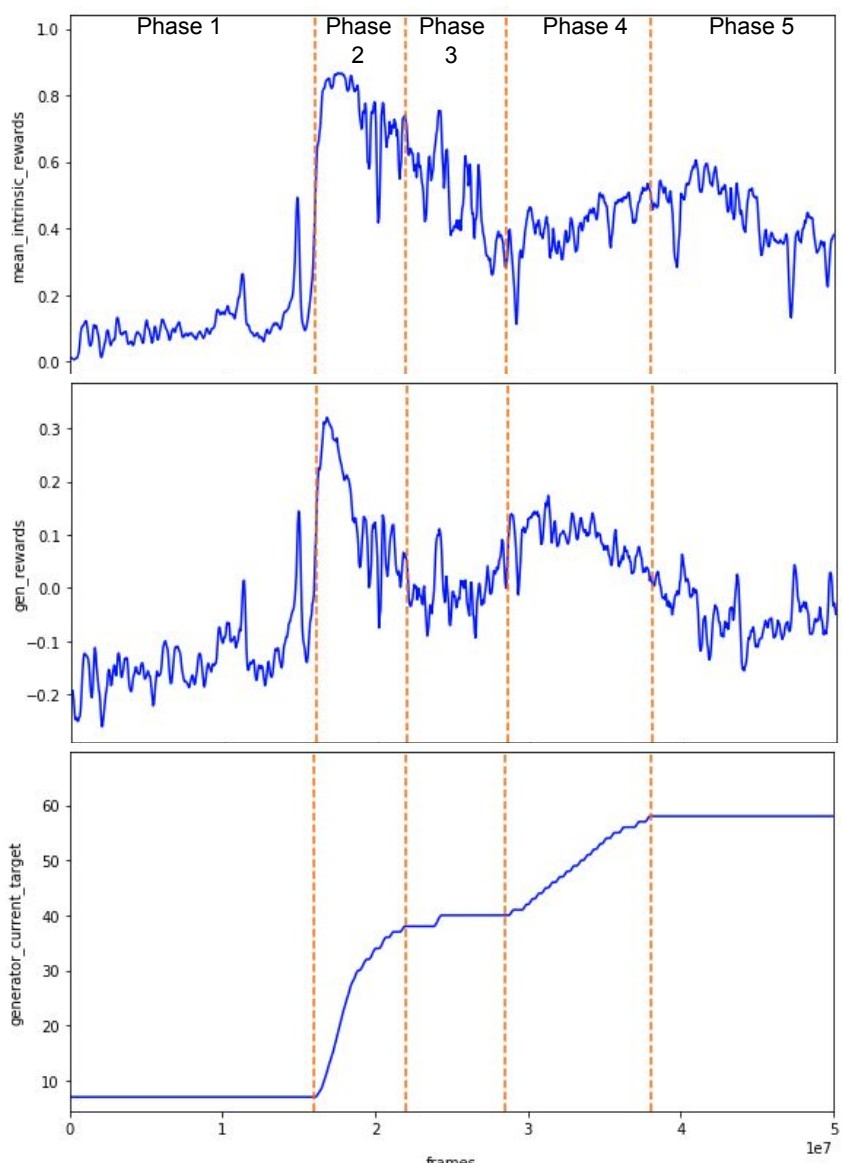

Figure 5: An example of a learning trajectory on **OMhard**, one of the most challenging environments. Despite the lack of extrinsic reward, the panels show the dynamics of the intrinsic rewards for the student (top panel), for the teacher (middle panel), and the difficulty of the goals captured as $t^*$ (bottom panel).

**Phase 4**: The student becomes too capable again and the teacher is forced to increase the difficulty of the proposed goals. **Phase 5**: The difficulty reaches a state where it induces a new equilibrium in which the student is unable to reach the goals and forced to improve its student.

AMIGO generates diverse and complex learning dynamics that lead to constant improvements of the agent's policy. In some phases, both components benefit from learning in the environment (as is the case during the first phase), while some phases are completely adversarial (fourth phase), and some phases require more exploration from both components (i.e. third and fifth phases).

Figure 3, presented in Section 4.4, further exemplifies a typical curriculum in which the teacher learns to propose increasingly harder goals. The examples show some of the typical goals proposed at different learning phases. First, the teacher proposes nearby goals. After some training, it learns to propose goals that involve traversing rooms and opening doors. Eventually, the teacher proposes

goals which involve interacting with different objects. Despite the increasing capacity of the agent to interact with the environment, **OMhard** remains a challenging task and AMIGo learns to solve it in only one of the five runs.

## F    GOAL EXAMPLES

Figure 6 shows examples of goals proposed by the agent during different stages of learning. Typically, in early stages the teacher learns to propose easy nearby goals. As learning progresses it is incentivized to proposed farther away goals that often involve traversing rooms and opening doors. Finally, in later stages the agent often learns to propose goals that involve removing obstacles and interacting with objects. We often observe this before the policy achieves any extrinsic reward.

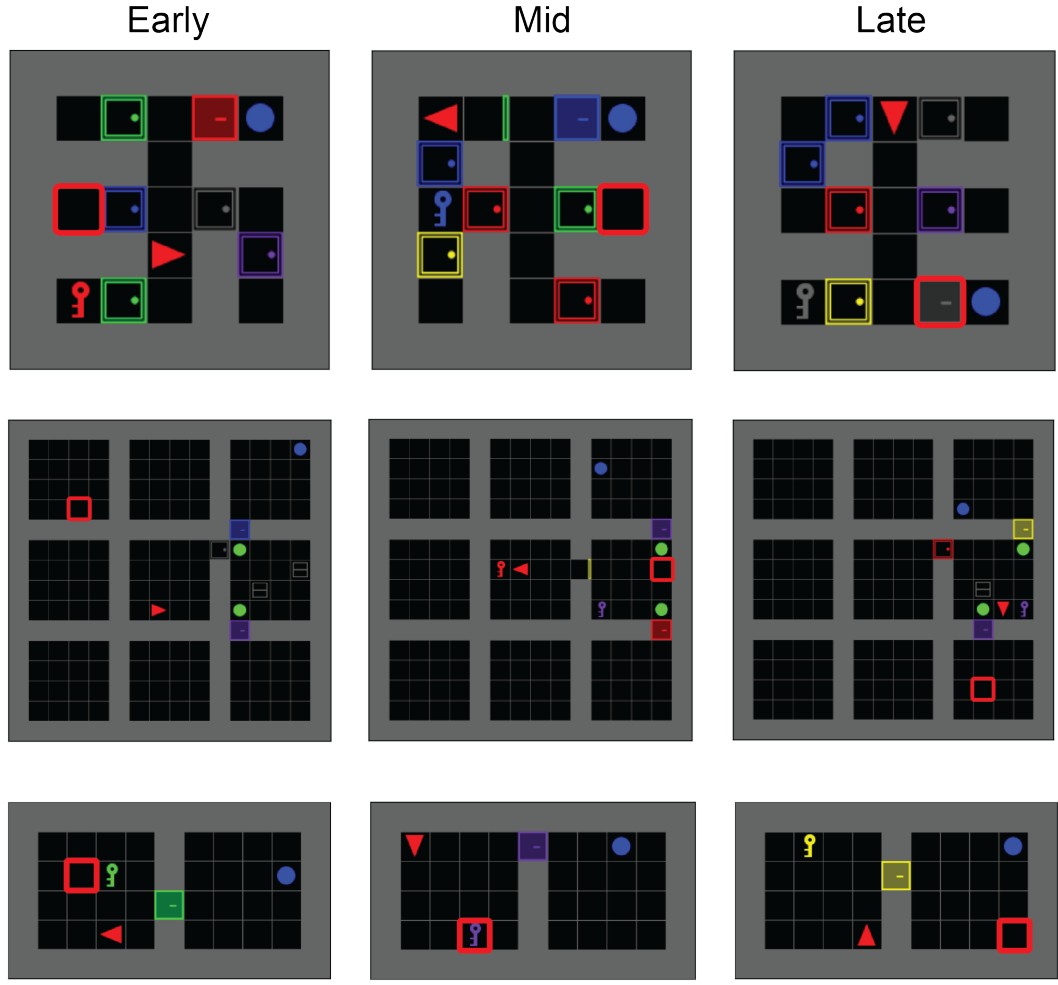

Figure 6: Some examples of goals during early, mid, and late stages of learning (examples for **KCmedium**, **OMhard** and **OMmedium** are first, second, and third rows respectively). The red triangle is the agent, the red square is the goal proposed by the teacher, and the blue ball is the extrinsic goal.

