# OpenReview forum: "Learning with AMIGo: Adversarially Motivated Intrinsic Goals"
_ICLR.cc/2021/Conference — ICLR 2021 Poster_

### Official Review · AnonReviewer2 · 2020-10-13
**Goal Generation for Directed Exploration**

**Rating:** 7
**Confidence:** 4

**Review:**

## After Author Response

The author response for most of the points of confusion and weaker sections of the draft are satisfactory. I thank the authors for incorporating some of the feedback during this discussion. I am still on the fence about the generality of this idea, and how much the teacher objective is specific to the minigrid tasks. But overall I find this idea interesting and believe it adds value to the field.

## Summary of the paper:
This paper proposes a goal-"generating" module (called the teacher) that will propose goals in procedurally generated grid world domains. The way they train this teacher is through policy gradient updates, with the teacher getting positive rewards if the agent achieves the provided goal but takes longer than a threshold to reach it, and negative rewards if the agent is unable to reach the goal or reaches it too quickly.

The threshold is gradually increased based on the agent's capabilities, allowing for more and more difficult goals.

Both the agent and the teacher additionally get rewarded for solving the external task. So the teacher is directed to provide goals that will also solve the given task, and the agent also learns to not just solve the given goal.
Other auxiliary objectives for the teacher are entropy regularization for diverse goals, and episode boundary awareness.



## Strengths:

+ This paper presents a clear reward based mechanism to train the teacher. It also presents a reasonable answer to the question of how a difficulty threshold should be set for the teacher. It provides evidence that this goal-generation approach helps in directed exploration, is useful for general RL tasks (beyond goal-based RL), and speeds up solving of a variety of hard grid world environments.

+ Figure 3 shows the changing thresholds and the subsequently generated targets for the agent, which is a good experiment to show that this increasing threshold mechanism works as intended.

+ The ablations in the appendix are illustrative and show the different considerations and how different components of the system interact.


## Weak Points:

- The suggestion that the goal generation is adversarial seems slightly strong, with not much evidence to show that the generated goals are adversarial to the agent. The goals could also be generated progressively farther and farther due to the moving threshold.

- The way the problem is set up and communicated makes it seem oriented towards grid-world problems exclusively. There is no definition of what a goal is. In fact, the problem setup itself is communicated rather vaguely, with terms being introduced as required rather than the entire setup being introduced clearly up front. Setting up the problem more clearly, with respect to RL literature (maybe refer to the Sutton and Barto book or alternative setups), being clear about what a goal is and how that goal is being used in the context of the grid world would make the paper clearer to read and understand.

- It is also unclear whether the teacher rewards are discounted according to the number of steps taken by the agent, or if it operates at a lower temporal frequency. The entire teacher policy training regimen needs more explanation and clarity.

- Given that the goal-generation here is to give an (x,y) location to travel to, it is unclear why an approach like GoalGAN cannot be adapted to this domain.

- An additional question to ask would be whether the approach of Zhang et al can be adapted as a baseline to compare to this work. I can understand if it is not applicable, since that approach is specific to goal-conditioned RL, but would like some clarity on it.

- When explaining the type of goals, the authors mention that this "includes picking up keys, opening doors and dropping objects onto empty tiles". How is an agent supposed to know whether it is supposed to move on to a square or drop an object there. The teacher does not seem to communicate what kind of goal is being generated. If the kind of goal is not communicated, wouldn't the agent do the simplest thing possible, and just move over the square, instead of figuring out that it should have dropped an object at the location?

- The auxiliary task of "Episode Boundary Awareness" is a little unclear. My understanding is that the teacher is rewarded for selecting goals where the object type at that location changes if the episode changes. An illustrative example or clearer language is necessary to understand this additional objective and how it is helping the agent. This is very important as it seems that this auxiliary reward is essential in getting the teacher to learn goals that are useful for eventual agent success (as evidenced by the ablations in Table 6). In fact, I am curious as to why the performance of the system if this particular reward is removed drops drastically and is similar to the extrinsic reward being removed.

- The main weakness of the approach for me is that it is unclear how this system scales to other RL problems. Since there is no definition of what a goal is considered to be, and no other examples given except grid worlds, it is hard to see what a broad application of this idea would be.

- The learning curves (Figure 4) seem to show that the training with AMIGo is pretty high variance, compared to the baselines. Is there an explanation for this variance? Example, performance on OMmedium drops to 0 pretty regularly.


## Additional Comments:

The related work section needs some additional work. While the authors present a broad array of related work, there are some suggestions to round off this section.

For intrinsic motivation, the authors should include work by Andy Barto ("Intrinsic Motivation and Reinforcement Learning", 2013) and other related work, as well as the optimal rewards framework by Satinder Singh's group ("Where do rewards come from?") along with work by Jonathan Sorg (ICML 2010) and more recently work by Zeyu Zheng ("On Learning Intrinsic Rewards for Policy Gradient Methods"). These views of intrinsic motivation as a vehicle for more than exploration is something that is not very clear in the current draft, even though the authors do present some alternative work like empowerment.

For curriculum learning, there are recent surveys that on curriculum learning for RL ("Curriculum Learning for Reinforcement Learning Domains: A Framework and Survey", Narvekar et al.; "Automatic Curriculum Learning For Deep RL: A Short Survey", Portelas et al.) that should be referred to when explaining curriculum generation in the context of RL.

## Conclusion

Overall I would like to see clearer communication of some of the ideas as well as some explanation from the authors that will give me confidence that this idea is more broadly applicable.

---

> ### Author Response · Authors · 2020-11-14
> **Some answers to your questions and comments (part 1)**
>
> We appreciate your constructive comments and detailed review which will help us to improve and clarify the paper. We will respond to your points and will improve our communication in the paper with the hope we can convince you our idea is broadly applicable to the point where you would consider improving your assessment.
>
> ### On the adversarial nature of AMIGo
> The loss function of the generator is partially adversarial by construction: the teacher is rewarded for proposing goals which are difficult for the student policy. We further qualify this as being “constructively adversarial”, because such goals must also be achievable, and thus not too hard. Empirically, learning trajectories (e.g. Figure 4) often show phases where the loss of the teacher decreases while that of the policy increases, demonstrating some notion of opposition in their respective objectives. Ultimately, you are right that the equilibrium condition is for the teacher to converge on proposing extrinsic goals once they are achievable, so in this sense it ceases to be adversarial at the end of training, but again, this is what motivates our terminology regarding “constructively adversarial”, which is used throughout the paper. We hope you agree with this characterization.
>
> ### Clarification regarding Goals
> AMIGo is applicable to a wide range of goal types, although whether it works or not is dependent on the implementation and modelling choices, and is thus the subject of further experiments. We intentionally underspecify the notion of goal at play to encourage such exploration. However, in our experimental setting, used to prove the concept for this general learning process, a goal is formally defined as a change in the observation on a tile, as specified by an x,y coordinate. We will add that statement up front along with a better clarification.
>
> We want to emphasize that AMIGo is applicable beyond grid worlds, although these are a good setting to prove the concept of this research direction, in line with many papers in the related literature. Specifically, the MiniGrid task suite was used because it offers sufficient complexity and range of difficulty and has been a popular benchmark in the related literature, e.g. in  [Igl et al.’s (NeurIPS 2019) paper on SNI](https://arxiv.org/abs/1910.12911), [Loynd et al.’s (ICML 2020) paper on WMG](https://proceedings.icml.cc/paper/2020/hash/5cf21ce30208cfffaa832c6e44bb567d-Abstract.html), or [Raileanu and Rocktäschel’s (ICLR 2020) paper on RIDE](https://iclr.cc/virtual_2020/poster_rkg-TJBFPB.html). Going beyond grid worlds, e.g., by first running semantic segmentation pipelines in photorealistic 3D environments and then defining intrinsic reward procedures around changes to detected objects would be an interesting research avenue, but outside of the scope of the paper. The overall idea and algorithm behind AMIGo is not grid world specific — only our particular choice of a goal proposal distribution is.
>
> We agree our choice of words was perhaps a bit unprecise when we say "includes picking up keys, opening doors and dropping objects onto empty tiles". We will formally introduce and clarify that those are among the different available actions for the agent to achieve a goal, namely a change in the (x,y) coordinate. In some cases in our setting, moving over a square is not the simplest thing possible, such as when an obstacle can be removed or a door opened. Similarly, in other tasks and environments it could be easier to affect a cell by throwing something at it than by reaching it. In our view, this is one way in which our goal definition can be achieved more generally.
>
> We hope these clarifications can convince you our setup is not oriented towards grid world problems exclusively.
>
> ### Clarification regarding the Teacher Training
> The teacher’s rewards are not discounted according to the number of steps taken by the agent. More specifically, the teacher receives reward after several steps depending on whether the agent reached a goal, so it does operate at a different temporal frequency. We will further clarify this on section 3.2:
> We did experiment with a fixed threshold, and found that the loss function alone was sufficient to induce harder goals and curriculum learning. We omitted to mention this due to lack of space and because fixed thresholds have to be carefully fixed for each environment (depending on its size and complexity), while the single moving heuristic is general and applies to all environments. We will explain this more clearly in the updated draft.
> Episode Boundary Awareness: Your understanding is correct and we will describe what we mean with an example.
> Observed Higher Variance in some Environments: We think the observed higher variance is due to the fact the learning dynamics of the two modules are coupled. Similar remarks have been made about GANs which can be difficult to train. In OMmedium, we did observe that the training stabilized after 27M steps for all runs.

---

> > ### Comment · AnonReviewer2 · 2020-11-18
> > **Comments on Author Response**
> >
> > Thank you for the detailed response. Some follow-up below.
> >
> > -  The point the authors make about the threshold and the teacher objective working in tandem to select goals which are at the edge of the agent's abilities is well made. I am leaning towards accepting the "constructively adversarial" qualification. One more question I have about the empirical behavior of this setup is whether the authors observe a "mode collapse" sort of problem. In particular, does the teacher end up proposing goals in one particular direction, and keep increasing the threshold there? Or does the exploration happen uniformly in all directions? Is this a challenge in training the teacher?
> > - While I am not disparaging the use of the MiniGrid tasks, nor the specific way the goal space is set up in these experiments, my point in highlighting the problem setup is so that this idea can be considered more generally. When the problem setup is not defined clearly, it's applicability to problems beyond the experimental settings becomes less clear. I would encourage the authors to setup the problem more clearly as suggested, and then describe the goal space and what it means to achieve a goal more concretely in the context of the MiniGrid tasks. That would make the paper more readable and more broadly applicable as well.
> > - Clarification regarding the teacher training will be appreciated, though it might require more discussion.
> > - Comparison of difficulty in training of the teacher and GANs is unclear. As the authors point out above the training objective is not adversarial in the sense GANs are adversarial. Why then do you see instability in training? Is it due to the hyperparameter of how quickly you move the threshold?
> > - I agree that AMIGo deals with a changing procedural environment more gracefully than GoalGAN does. My comment was more along the lines of why GoalGAN was not compared to in experiments, if the goal that is to be generated is an (x, y) coordinate, rather than a state the agent needs to get to. I am not convinced that GoalGAN cannot be used as a baseline for comparison in this setting.
> >
> > In conclusion, I am happy with the author response, and am willing to revise my score upwards if the authors update the draft to incorporate the feedback from the reviewers.

---

> > > ### Author Response · Authors · 2020-11-19
> > > **Some answers to your follow up comments and questions (part 1)**
> > >
> > > Thank you for your quick response and insightful comments. As you can see in our [summary of changes post](https://openreview.net/forum?id=ETBc_MIMgoX&noteId=JMde7SZ75tT) (and in the updated paper itself) we have incorporated your feedback from the first review and your follow up comments into the paper. We trust you will find everything to your satisfaction in these amendments, and in the responses below, and feel secure in your support for our paper.
> > >
> > > ### On “mode collapse”
> > > We did not observe a “mode collapse” problem in our experiments. Note that, as discussed in Section 3.4, the teacher is also rewarded when the student reaches the extrinsic goal. This incentivizes the teacher to propose goals that are useful for reaching the extrinsic goal. Thus, generating goals in a “single direction” would not maximize the teacher’s objective. Moreover, the notion of a “direction” is not well defined in environments such as MiniGrid where it is not just about walking in a certain direction. Instead, a goal (e.g. reach a tile behind the door) might require very diverse skills (e.g. moving away a box or finding a key), so the goals posed by the teacher likely require diverse skill sets (as in e.g. ObstructedMaze). We demonstrate the importance of this aspect of the teacher reward with regard to avoiding a form of degeneracy in the teacher-student interaction, in the ablation study described in Appendix D and Table 6. We agree that further qualitative analysis of the proposed goals would be interesting, but we believe this is better reserved for follow up work on richer environments such as [NetHack (Kuttler et al. NeurIPS 2020)](https://arxiv.org/abs/2006.13760).
> > >
> > > ### On the problem setup
> > > Thank you for this suggestion — we agree it is important to be very clear about the problem setup we consider. We added further details about our setting in Section 3.3 of the revised draft and we hope the generality of our setup is now more apparent. To summarize, as a proof of concept for AMIGo-style training, we define a goal (x, y) as a change in the content of the tile at the (x, y) coordinates. This definition induces a rich set of behaviors as different skills are required such as finding keys, moving away obstacles etc. Figure 3 also shows a few examples of the proposed goals in a MiniGrid environment, which we hope helps ground our instantiation of the goal space.
> > >
> > > ### On the teacher training
> > > Thank you for pointing this out. We added more details about training the teacher in Section 3.2, as well as in Section 3.4 in the paragraph on Episode Boundary Awareness.
> > >
> > > ### On training instability
> > > The question of instability in training is a complex one to address. First, it is important to note that the teacher is also trained via policy gradient methods, and thus there is high variance at the core of its training regime.
> > >
> > > Beyond that, our current hypothesis is that any additional instability is due to the semi-adversarial way of training the teacher and student simultaneously. The student is trained in a regime where the rewards are non-stationary (due to the fact that the intrinsic reward changes as the teacher is updated), so after learning to obtain some extrinsic reward on the task of interest (while still not fully solving it 100% of the time), the threshold is moved so the teacher starts proposing different goals which the student cannot achieve yet. Thus, the student stops receiving reward for its good behavior and since its skills are not yet enough to obtain consistent extrinsic reward, it might update its policy in the wrong direction so its performance on the task drops. After some more training (and goal exploration), the teacher likely figures out those goals are too difficult for the agent and starts proposing easier goals that the student can achieve and start learning useful skills again, which eventually lead to another increase in the extrinsic reward.
> > >
> > > However, we do not think this is necessarily a problem with how quickly we move the threshold. We believe this behavior might be inherent to training a student and teacher simultaneously with this type of “constructively-adversarial” objectives.

---

> > > ### Author Response · Authors · 2020-11-19
> > > **Some answers to your follow up comments and questions (part 2)**
> > >
> > > _(following [part 1 of our response to your follow-up questions](https://openreview.net/forum?id=ETBc_MIMgoX&noteId=06CvNz6w1w0)...)_
> > >
> > > ### On GoalGAN
> > > While GoalGAN is an interesting related approach, we did not compare AMIGo against it because it was designed for a different setting (namely a continuous control singleton environment rather than a discrete procedurally generated one), and thus it is not trivial to adapt to our setting. Adapting GoalGAN to our setting would require significant changes such as adding the initial state as an input, as well as modifying the generator’s output, which already departs from the original GoalGAN (so a direct comparison with the authors’ implementation is infeasible). This decision was also informed by the conclusions in Racaniere et al. (2019) who were unable to adapt GoalGAN to an environment that changes with each episode because of the challenges in training GANs conditioned on high-dimensional inputs (which would be the case for our environments). Other differences between GoalGAN and AMIGo such as the need for a memory buffer as well as more computational resources would further complicate judging the two methods in a fair way.
> > >
> > > ### Summary
> > > We are grateful for your thorough feedback and engaged discussion which has helped improve the quality of our work. We look forward to hearing from you if you have further comments, and hope for your strong support for the paper.

---

> ### Author Response · Authors · 2020-11-14
> **Some answers to your questions and comments (part 2)**
>
> (following from [part 1 of our response](https://openreview.net/forum?id=ETBc_MIMgoX&noteId=0H9w0X0FfK)...)
>
> ### Additional Work on Related Section
> We appreciate your pointers to the alternative views on intrinsic motivation as a vehicle for more than exploration, and to the surveys on curriculum learning. With the additional 1-page we will be able to include and briefly put in context our model in relation to this work.
>
> The work of Zhang is concurrent to ours. While it was proposed in the context of goal-conditioned RL for continuous control, we agree that it would be interesting to compare it with AMIGo. However, we believe this is outside the scope of this paper.
> GoalGAN: Please see [our reply to Reviewer 1 regarding GoalGAN](https://openreview.net/forum?id=ETBc_MIMgoX&noteId=QG_ZJLeK5Q8), whose generator is not conditioned on the initial observation and thus is not easily adapted to environments that change from episode to episode.
>
> ### Summary
> Your concerns have helped us understand how we can more clearly communicate our contributions. We hope these clarifications can convince you of the general applicability of our idea, and that you will consider increasing your support for the paper. We would be grateful if you can let us know of any outstanding concerns preventing you from raising your score, or can indicate to us a way of making further improvements.

---

### Official Review · AnonReviewer4 · 2020-10-28
**Cute idea, but may be difficult to generalize and scale up**

**Rating:** 6
**Confidence:** 4

**Review:**

This paper introduces a teacher agent in reinforcement learning framework. The teacher's responsible for setting goal for the student agent (the "main" agent). The teacher acts in adversarial way (but also somewhat cooperative) in setting the goal. That is, the goal should be achievable by the student, yet the goal should not be too simple that the student stops learning towards harder goals, and eventually achieve the original goal from the environment. Experiments show proposed method is able to learn okay performance for hard environments, even though most other methods fail to score at all.

Pros:
I think this is a very interesting approach and has great potential. The idea is natural and experiments make a lot of sense. This should partially credit to clear writing and examples. The authors provide only one set of results in main paper, but have more ablation studies and variants of reward forms. Those results will be helpful to provide more insight into how well the agent handles the task.

Cons:
My major concern with proposed method is that it seems limited to particular applications, such as grid world. As the authors also pointed out, even proposing goals in this paper's setting is a challenging problem. For now, I don't see an automatic way to proposing goals for other reinforcement learning tasks, such as Atari games and Go. In those tasks, I feel teacher's goals may need to be designed incorporating human's understanding of the tasks (i.e. capture a stone in Go). This makes me feel the proposed method is not very generalizable, but I am open to change my opinion if authors provide a good answer.

If for those tasks, a human-generated set of goals are needed by teacher agent, then how do we correctly attribute the performance gain we saw in this paper? That is, are the gains over baseline methods from 1) we have a better training framework, such as curriculum learning, or 2) the goals are set in a way that incorporate our understanding of the environment?

My second concern is about scalability of proposed method. For some applications, it may be unnecessary to have a teacher that sets intermediate goals (e.g. AlphaGo -> AlphaZero). How does sample efficiency for AMIGo compared to baselines? I saw results related to number of steps for AMIGo alone, but not baseline methods. Conceptually if a teacher sets many intermediate goals, it's likely that the student will train more number of steps and possible to achieve better performance. However if this procedure requires 10x more number of steps to achieve on-par performance for easy/medium level tasks, then it will be important to acknowledge that. When will the student need a teacher vs. w/o a teacher seems also worthy studying.

---

> ### Author Response · Authors · 2020-11-13
> **Some answers to your questions and comments**
>
> We thank the reviewer for their comments. It seems that the two main concerns the reviewer has have to do with the applicability of AMIGo to different settings or with different goal types, and the scalability and sample efficiency of the method. We answer both of these concerns below.
>
> ### Limitations of AMIGo to Grid Worlds, and other Goal Forms
> The review is concerned that the choice of intrinsic goal specifications limits the applicability of AMIGo to grid worlds. We should be clear about the aim and claim of this paper: it is to demonstrate that this general “constructively adversarial” form of teacher-student training works as an exploration mechanism. We are transparent about the general requirements of this method throughout section 3: namely that there exists a form of intrinsic goal the student policy can condition on, that there exists a (learned or hard-coded) valuation function over these goals, and that there exists a measure of the student’s performance. These requirements do not restrict us to grid-worlds or any specific environment, but particular choices of goals and performance measures might. To prove the concept, we have focussed on a particular choice of goals (coordinates in a grid world, which gives us a valuation function for free) and performance (steps), in order to have an in-depth comparison with other exploration methods on a well-known exploration problem, MiniGrid. The MiniGrid task suite offers sufficient complexity and range of difficulty and has been a popular benchmark in the related literature, e.g. in  [Igl et al.’s (NeurIPS 2019) paper on SNI](https://arxiv.org/abs/1910.12911), [Loynd et al.’s (ICML 2020) paper on WMG](https://proceedings.icml.cc/paper/2020/hash/5cf21ce30208cfffaa832c6e44bb567d-Abstract.html), or [Raileanu and Rocktäschel’s (ICLR 2020) paper on RIDE](https://iclr.cc/virtual_2020/poster_rkg-TJBFPB.html).
>
> We cannot and do not claim that this method will work out-of-the-box in other domains, with other goal types, but explicitly leave this for further work, as pointed out at the end of Section 5. Nevertheless, we do not see a substantial difference between many tasks such as Atari and our setting in terms of requiring deeper human understanding of the environment. Many MDPs discount rewards based on the number of steps, and often difficulty can be correlated with the amount of effort or time needed to obtain a certain outcome, this could be sufficient for the teacher to learn to propose appropriately difficult goals.
>
> However, we agree with you that investigating other settings would be interesting, and present particular challenges, such as depending upon human knowledge to model the goal valuation function. There is prior work in this area incorporating aspects of inverse reinforcement learning, e.g. [Bahdanau _et al._ (ICLR 2019)](https://openreview.net/forum?id=H1xsSjC9Ym) where a language-conditional goal-verifier is learned jointly with a policy, based on a finite set of expert examples of goal states, all done on pixel input. But it is clear that this is ambitious and non-trivial research that must be the subject of further papers, rather than trying to cram all possible work and evaluation in this one (which has no fewer than 85 experiments). We hope you will agree this paper makes a significant enough contribution to both warrant publication, and enable such follow up work to be done.
>
> ### Scalability and sample efficiency
> The reviewer will be happy to hear that their concerns about scalability and sample efficiency are already addressed in the submitted version of the paper, namely in Appendix C, as pointed to from Section 4.4. In this appendix, we show that on the two easiest environments, **KCmedium** and **OMmedium**, agents need about 10 million steps to converge while on the other four more challenging environments, they need an order of 100 million steps to learn the tasks, showcasing AMIGo's contributions not just to solving the exploration problem, but also to improving the sample complexity of agent training. Naturally, for the harder environments, sample efficiency is not comparable since AMIGo is the only method obtaining non-zero returns. We hope this addresses your second main concern, and are open to hearing how we could make things clearer in the paper.
>
> ### Summary
> We believe that both the response above, and the existing plots and discussion from Appendix C, sufficiently address your concerns to the point where you’ll consider revising your assessment in support of the paper. To repeat our main point, the paper proves the concept for a new class of exploration methods, and thoroughly compares it to existing state of the art. We are confident that it does so, and unlocks further avenues of investigation, including some of those astutely suggested by the reviewer. We hope for your support in these matters, and are at your disposition to further answer any questions or concerns you may have.

---

### Official Review · AnonReviewer1 · 2020-10-29
**SoTA results on MiniGrid, similar ideas in previous methods, additional discussion about related works is needed.**

**Rating:** 6
**Confidence:** 4

**Review:**

This paper proposes a setter-solver or teacher-student scheme for training goal-conditioned agents (student/solver) in a discrete environment MiniGrid. In their method called AMIGo, the teacher takes the initial state of the student at the beginning of each episode as its input and proposes an intrinsic goal for the student to fetch, the student in each step takes an action and gets a reward combining both the extrinsic reward from the environment and the intrinsic reward, and the teacher gets a reward at the end of an episode or when the intrinsic goal is reached. The major idea is to make the goals proposed by the teacher staying at an appropriate/medium level of difficulty, i.e., fetchable for the student but not too easy. To do so, they use a threshold to define the teacher reward: they issue a positive(negative) reward to the teacher if the student spends more(less) steps than the threshold to reach the goal. And they adaptively increase the threshold if the student keeps successfully reaching the intrinsic goals. In experiments on five environments of MiniGrid, AMIGo outperforms five baselines with different types of intrinsic reward and set up new SoTA results on the studied tasks.

This paper is well-written and demonstrates a reasonable curriculum strategy by training a teacher model to propose goals in a discrete grid world (though largely simplified from realistic scenarios). The experiments show remarkable improvements over existing curiosity-driven or intrinsic reward-based methods and set up a new SoTA on those tasks.

My major concerns are the novelty of the idea and the generality of the proposed method.

(1) Though the authors discussed the difference of AMIGo to GoalGAN, in which the teacher is instead a GAN generating continuous goals, the main idea behind the two are very similar, i.e., to train a teacher model proposing goals of medium difficulty, and they both use threshold(s) on rewards for this purpose. Although I agree that they use different architectures for the teacher model (for producing different types of goals), and the GAN in GoalGAN does not take student's initial state as input since it is not dealing with procedurally-generated environments, these are natural choices adapted to the specific environments/tasks for testing these methods. The main contribution of this paper, from the perspective of proposing a novel curriculum generator, is incremental.

(2) Another related curriculum generating strategy is to balance the difficulty and diversity/representativeness of goals/tasks when proposing them to train the goal-conditioned student. It worth discussing the difference/advantage/limitation of the proposed method when compared to them. To name a few,

Portelas et al., "Teacher algorithms for curriculum learning of Deep RL in continuously parameterized environments", CoRL 2019.
Fang et al., "Curriculum-guided hindsight experience replay", NeurIPS 2019.

(3) The experiments only demonstrate the method in one type of environment. Although the MiniGrid is an appropriate environment to test the proposed curriculum generating method due to its simplicity, which allows the authors to rule out other factors and mainly focus on the quality of selected goals, it is not clear whether/how the proposed method can be easily generalized to other more practical and/or complicated environments/tasks such as robotic control, navigation, traffic, and gaming. More discussion on how to adapt AMIGo in those tasks can benefit researchers following this work.

(4) The student is a goal-conditioned policy and it can be conditioned on one goal per time. During training, when is the student conditioned on the extrinsic goal or the intrinsic goal? Is there a schedule of assigning which goal to the student?

(5) The student can also self-propose a goal by hindsight experience replay (HER) and this has been demonstrated to be effective in alleviating the sparse reward problem. It is interesting to see a discussion or comparison to HER methods.

---

> ### Author Response · Authors · 2020-11-13
> **Some answers to your questions and comments (part 1)**
>
> Thank you for your comments and questions, and for your support for the paper. We will respond to your points and answer your questions in the hope that our discussion will not only help us improve the paper, but also help us understand what outstanding issues (if any) stand between us and a stronger endorsement on your part after discussion.
>
> ### Novelty with regard to GoalGAN
> We respectfully but robustly disagree that novelty is an issue here, but are keen to make the point more clearly in the paper and hope you will help us improve on this axis. First, GoalGAN uses a discriminator to evaluate the difficulty of goals, as trained in an adversarial setting. In contrast, our teacher network is a “constructive adversary”, as its own objective forces it not only to model what is difficult for the student, but also what is feasible, and propose appropriate goals. Second, the fact that GoalGAN does not take the initial state into consideration can be considerably limiting and is not just an easy adaptation chosen because of the environment. For example, Racaniere et al. (2019) do compare against GoalGAN in the locomotion tasks but are unable to adapt it to their Alchemy environment which changes from episode to episode. Furthermore, to this point they argue that training generative models with non-trivial conditioning is challenging in general, and in this context, discovering the relevant environment and goal structures is difficult.
>
> This also yields fairly significant additional differences:
> 1. Their training is considerably more complex as it requires a three-step iterative procedure and depends on the coordination of three different modules (the generator, the discriminator, and the policy), while in our method the teacher and student are trained simultaneously.
> 2. GoalGAN requires a memory buffer of previous goals proposed. Maintaining and sampling this large memory can be very costly. In addition, as noted, previous goals can become unfeasible or nonsensical through time or if the environment changes.
> 3. Having to run the agent on each goal multiple times to get a label of whether it has an intermediate difficulty requires additional wallclock time and computation for the same number of agent steps.
>
> ### Relations to Other Curriculum Learning Strategies
> Thank you for flagging the work of Fang _et al._, and Portelas _et al._ We were not acquainted with these papers. We are happy to include a discussion of this work given the additional page afforded us for the final paper, and will do so during the discussion period. While there are fairly significant differences with regard to these papers and ours both in terms of to the methods and applications (e.g. Fang _et al._ focus on off-policy learning, whereas our method is primarily aimed at on-policy learning), you are right that further investigating the balance between difficulty and diversity of goals is worth investigating in further work. We are confident complementary gains are to be made, but hope you agree that further investigation of this is outside of the scope of the present paper.
>
> ### On the use of Minigrid
> We completely agree that the ubiquitous applicability of this method to other domains needs to be empirically tested, and we are careful not to make predictions here. We limit ourselves to stating what limiting assumptions are made in the general approach, and in our specific implementations, to demonstrate the _potential_ for application to other domains. Given the space constraints, we chose to focus on an in-depth evaluation and comparison on a single domain with a diversity of exploration tasks and settings, to prove the concept for this exploration method, in line with other work in the literature.
>
> On this point, we emphasise that despite its visual simplicity, the MiniGrid task suite offers sufficient complexity and range of difficulty and has been a popular benchmark in the related literature, e.g. in [Igl et al.’s (NeurIPS 2019) paper on SNI](https://arxiv.org/abs/1910.12911), [Loynd et al.’s (ICML 2020) paper on WMG](https://proceedings.icml.cc/paper/2020/hash/5cf21ce30208cfffaa832c6e44bb567d-Abstract.html), or [Raileanu and Rocktäschel’s (ICLR 2020) paper on RIDE](https://iclr.cc/virtual_2020/poster_rkg-TJBFPB.html).

---

> ### Author Response · Authors · 2020-11-13
> **Some answers to your questions and comments (part 2)**
>
> _(following from [part 1 of our response](https://openreview.net/forum?id=ETBc_MIMgoX&noteId=QG_ZJLeK5Q8)...)_
>
> ### On goal conditioning for the student
> In the minigrid task-suite proposed, the goal is part of the observation space, rather than provided by a separate input (as done for the intrinsic goal provided by the teacher). As such, the student conditions on both the intrinsic goal and extrinsic goal at all times, and is free to “decide” which to follow. Two points should be made here:
> 1. There is nothing about our approach which prevents it from being applied to cases where the student received the extrinsic goal from another input/modality (e.g. instruction following)
> 2. As pointed out at the end of Section 3.4, rewarding the teacher when the extrinsic goal is reached by the student incentivises the teacher to converge on proposing the extrinsic goal at the end of training, avoiding degenerate behaviour such as the student having to choose between a reachable extrinsic goal and an intrinsic goal which would potentially lure it away from completing the extrinsic task.
>
> ### Connection to HER
> While our method shares some similarities with HER in terms of having a policy conditioned on intrinsic goals, there are significant differences between the two methods. In the case of HER, there is no explicit incentive for the agent to explore beyond its current reach. In fact, HER is rewarded for all the states it visits which means that it is rewarded for easy-to-reach states even late in the training process. This reduces the agent’s incentive to further explore its environment. In contrast, our agent is constantly encouraged to visit new states via the teacher’s reward function. Thus, in the context of experimenting with exploration methods, these are not comparable. Naturally, the question of their complementarity is worth investigating in future work, but this is not the focus of this paper.
>
> ### Summary
> We hope to have suitably addressed your main concerns in this response to the point you would consider increasing your support for the paper. If this is not the case, we would be grateful if you could indicate where we fall short so that we may further improve the paper with your feedback.

---

### Official Review · AnonReviewer5 · 2020-11-06
**Intriguing, simple idea, but worries about experimental methods**

**Rating:** 7
**Confidence:** 3

**Review:**

The authors introduce AMIGo, an approach to curiosity in which an adversarial "teacher" agent proposes goals that the "student" agent attempts to achieve. The student obtains an intrinsic reward of +1 when it achieves the goal, and this augments the extrinsic reward additively. The teacher proposes a goal at the start of each episode, and whenever the student reaches a goal. The teacher is positively rewarded if the student achieves the goal after some threshold t*, and negatively rewarded otherwise. t* is incremented at fixed intervals. They then demonstrate that this outperforms a variety of suitable baselines.

The algorithm proposed has great strength in its simplicity. It does bear a considerable similarity to Sukhbaatar et al 2017 (their ASP baseline), but AMIGo appears to be more flexible, and they put forth evidence that it considerably outperforms ASP on their benchmarks. AMIGo is something that could be adapted very easily to many other settings, and it is a quite agnostic framework. The experimental results do show considerable advantages over baselines.

I am, however, concerned about how the evaluations were done. At the very least, I find that the main text fails to describe what seem to be important caveats that can only be found in the appendix. In particular:
-For baselines, suitable hyperparameters were found using a subset of tasks (tasks thought to be easier), and, if positive results were achieved, the same hyperparameters were run on the rest of the tasks. If scores of zero were obtained for the initial subset, the baselines were not run at all and were assumed to score zero.
-As noted in the main text, some of the baselines were developed for partially observed environments with an egocentric view, so they let the baselines run in both modes (with algorithms with multiple modules, they varied this independently for both). While this might seem like simply giving the baselines more chances to succeed, there is a problem when this is coupled with the first practice of assuming zero scores. Some of these baselines obtained no score in the fully observed setting on the "easy" task subset, and hence were not run on the hard environments. They did better in partially observed modes, and hence obtained their hard environment scores in these partially observed modes. These partially observed modes are potentially much harder!

While one might be able to make some claim of the form "if they can't do the easy task, they can't do the hard," I am worried that these choices, used together, really obfuscate the performance differential. Note the quite high score variance that AMIGo achieves on the harder environments. Given such high variance, models which were simply not run in the fully observed setting could have actually performed better were they given the chance (and also perhaps needed further hyperparameter tuning). At the very least, I think this is a really unclear way of benchmarking things and should be corrected so that we do not have these statistical worries.

As a result, I cannot at this time recommend acceptance, though I am certainly open to being persuaded that my worries are unfounded. I could have missed something about their experimental methods, or environmental particulars could somehow make the above treatments perfectly reasonable.

A more minor concern: I do wonder about the extent to which this method applies in other settings. Is this time-threshold reward method applicable to a broad array of settings, or is this particularly useful in minigrid and similar sorts of maze  navigations? Are there situations in which time-to-complete is not a good proxy for difficulty? Is it important that the goals be represented in some very compact way, as they are for these experiments? These are questions that would, I think, need additional experiments. The authors did a considerable amount of evaluation, so I hesitate to simply ask for more, but I do wonder if this approach is particularly suited to a narrow range of environments.

I am also curious as to why the ASP baseline did so poorly. Its method seems to be quite similar, so more commentary about what makes one fail badly while the other succeeds would be really useful!

*Updated score based on below discussion*


*Updated score again based on below discussion*

---

> ### Author Response · Authors · 2020-11-12
> **Clarifications regarding experimental protocol and results**
>
> Hello, and thank you for your detailed review. It sounds like you like the paper and the method, especially regarding the potential to apply it to a wider range of RL settings. We agree, and would be significantly empowered to pursue further research in this area if we can convince you the paper proves the concept and is worthy of publication!
>
> ### Experimental Protocol
> Our understanding is that the main thing preventing you from wholeheartedly supporting acceptance is the concerns you outline about the experimental method. We aim to address those during this discussion, both through clarification and through running the outlying experiments you wish us to verify. We hope that this will not only address your concerns, but give you the comfort you need to fully support the paper with your score.
>
> We believe we performed an extremely thorough experimental analysis and comparison of AMIGo against comparable benchmarks, (actually) running 85 experiments (each with 5 seeds), each of which requires non-trivial computational resources (1-2 dedicated GPU days per experimental run, per seed!). Available resources barely permitted this, and because individual experimental runs showed that no method would solve harder experiments if flatlining (constant zero return) on easier tasks, it seemed fair and intuitive to avoid running methods on harder tasks if they cannot train *at all* on easier tasks. We were as transparent as we could be about this when reporting results, and hope you will understand this was not an attempt at obfuscation.
>
> However, for the avoidance of doubt, **we ~will~ have run all the remaining experiments projected to flatline, following the same experimental protocol used for the reported experiments, to confirm our projections**. We confirm that our results stand in [this post](https://openreview.net/forum?id=ETBc_MIMgoX&noteId=4dnHW4XDd_M). Experiment code will be open-sourced so the community can check our results for maximum transparency.
>
> ### Choice of Threshold Cost
> The reviewer astutely suggests that alternatives to the number of steps could be used as a cost, for the purpose of training the teacher. We do emphasise in Section 6.2 that there are several options available for measuring policy performance in this context, and that step count was a simple choice for the purpose of our experiments. In more complex settings, other measurable costs might be suitable, for example power consumption during the operation of a robotic arm in a continuous control setting. You are correct that to properly evaluate the general framework in more general settings, further experiments are required. However, to prove the concept for this method, the MiniGrid task suite offered sufficient complexity and range of difficulty and has been a popular benchmark in the related literature, e.g. in [Igl et al.’s (NeurIPS 2019) paper on SNI](https://arxiv.org/abs/1910.12911), [Loynd et al.’s (ICML 2020) paper on WMG](https://proceedings.icml.cc/paper/2020/hash/5cf21ce30208cfffaa832c6e44bb567d-Abstract.html), or [Raileanu and Rocktäschel’s (ICLR 2020) paper on RIDE](https://iclr.cc/virtual_2020/poster_rkg-TJBFPB.html).
>
> We hope you will agree that the extensive experiments done here at least show the robustness of our method within a difficult class of exploration problems, and that future work seeking to adapt it to different contexts warrants further research, to be incorporated into future papers.
>
> ### ASP Baseline
> We used the authors' own [PyTorch implementation of ASP](https://github.com/tesatory/hsp), and did a thorough hyperparameter sweep, so we are confident the empirical results are a fair comparison. We are unsure how to explain this result, and are not certain it is our place to proffer negative analysis of others’ work, but we should clarify that ASP is not that similar to AMIGo. ASP requires two policies acting, and can only propose goals which have been reached by one of them. Furthermore, ASP requires reversible and resettable environments, which is a pretty strong assumption that AMIGo does not make. We hope these crucial differences go some way towards giving you an intuition as to why there is a difference in performance, and wider potential for application. We are happy to add something about this in the final paper if you think it essential, as there is space for it with the additional page, but we initially decided not to focus on this too much to avoid the impression of talking down other researcher’s methods, as the numbers spoke for themselves.
>
> ### Summary
> We hope the clarifications above, paired with the experimental confirmation of our projections for the harder environments (which we expect to provide sometime next week), are sufficient for you to reconsider your assessment of the paper. If you have any outstanding concerns, please don’t hesitate to let us know so that we can discuss them, and properly understand what—if anything—stands between us and a strong recommendation for acceptance.

---

> > ### Author Response · Authors · 2020-11-15
> > **Confirmation of results**
> >
> > We have run all experiments that were projected in Tables 2–5, including additional seeds for ASP where we had only run one seed, and we can confirm that the results stand: all runs that were projected to have 0.00 MER have 0.00 MER. We will update the paper accordingly.
> >
> > We hope that, with this further experimental validation, you are satisfied that the experimental results are robust and that the paper merits your support. If you have any outstanding questions, please do not hesitate to let us know. We thank you again for your feedback, and for encouraging us to check these results empirically.

---

> > > ### Comment · AnonReviewer5 · 2020-11-15
> > > **Thanks for running!**
> > >
> > > Ok, this is good to hear. And yes, this methodology was my main concern by far -- I'll plan to update my score. Before doing so, I was hoping the authors could interpret these results a little, for me.
> > >
> > > In particular, what's striking about this result is that all of these baselines seem to utterly fail in the fully observed setting, and most seem to only gain any traction when both the policy and the intrinsic motivation-computing components are made partially observed and agent-centric. It's difficult for me to think of these modifications as comparable to the fully observed setting: the agent-centric transformation is a simple one (but one that can matter a good deal especially when computing some of these intrinsic motivation signals), and the partial observability can make the task *much* harder (albeit potentially leading to some boosts, e.g. due to the input space being smaller and less varied).
> > >
> > > That these baselines fail in the fully observed setting, then, seems like a really important part of interpreting these results. So, why do we think the baselines completely failed in the fully observed setting? Was it the agent-centric aspect of things? Or the partial observability? Or a combination of both? I hesitate to demand you run more experiments, but varying only one at a time could be informative. E.g. if we think about RND or RIDE, it seems plausible that the intrinsic motivation term might really benefit from the agent-centric view (if I understand these environments correctly, not a lot changes in the view used in the fully observed setting), but the effect of making things more partially observed seems unlikely to help. And do we think there is something particular to these sorts of tasks?

---

> > > > ### Author Response · Authors · 2020-11-17
> > > > **Further response to your comments and questions (part 1)**
> > > >
> > > > Thank you for reading and replying to our first response so quickly. We are happy to hear you are motivated to reconsider your assessment now our experimental results are verified, and have some additional questions. We will answer them to the best of our ability, as we are eager for you to be as comfortable as possible in giving your (hopefully strong) support for the paper.
> > > >
> > > > ### On the failure of baselines in the fully observed setting
> > > > It is an interesting observation that some baselines fare particularly poorly in the fully observed setting, and we are happy to speculate about this here, but we first must make the point that this is an orthogonal question to the point studied in the paper (namely, does AMIGo work better than alternatives?). In fact, to ensure that the baselines were compared against in their _best_ setting, we remind the reviewer that the baseline/benchmark results of Table 1 are the pointwise max of the results from tables 2-5 of the appendix, so each baseline _has_ been evaluated in all possible observation settings for that baseline.
> > > >
> > > > ### On using partial or full observations for the intrinsic rewards (e.g. RND / RIDE / ICM)
> > > > Let us first consider the question of whether it is better to use partial or full observations for computing the intrinsic reward. If the intrinsic reward uses a full view, the agent might be intrinsically rewarded with the same reward for states (i.e. full obs) that are very different, so it might make learning more inefficient because those states might have different optimal actions and, it is difficult to learn an intrinsic reward using the full obs. You have clearly surmised this, as you ask **“if we think about RND or RIDE, it seems plausible that the intrinsic motivation term might really benefit from the agent-centric view”**. This is precisely the case: the reason why it makes more sense to use a partial view for computing the intrinsic rewards in the case of RND, RIDE, and ICM, is that it better represents what is new / changing or surprising in the environment. If you use the full view, in most cases, the only thing that changes across different states of the same episode is the position of the agent (and perhaps some property of an object) but this is likely to be interpreted as a very small change in the state representation given that only a few of the inputs are changing (and the input is high dimensional in the full view). This leads to uniform intrinsic rewards across the different states which makes it difficult to distinguish "novel / interesting / surprising" states from less "interesting" ones, which in turn makes it harder to learn in such sparse reward environments (in practice, it reduces the setting to one without intrinsic rewards).
> > > >
> > > > In the case of RIDE, if you use the full observation, you will end up with similar (and very small) changes in your state representation no matter what action you take, which translates into uniform and small intrinsic rewards for all states. In the case of RND, if you use the full observation, you will probably end up with either very high rewards for all states in an episode if that layout has never been seen before or with very low rewards for all states in an episode if that layout (or a similar one) has been seen before. This is because the things the agent can control (i.e. its location) and change in an episode are a small subset of the entire input so the layout will have a much higher weight than the local observation around the agent. A similar story applies to ICM, where what you care about is the effect the agent can have on the state which is always local in these environments (and in general).
> > > >
> > > > Regarding whether these observations are particular to MiniGrid or more general, that is a good question. We believe these insights might apply more broadly since in many cases the things the agent can control / change in the environment (on which many of the intrinsic rewards are based) are local. In particular, if you have high-dimensional observations, they will contain a lot of information that is irrelevant for computing the intrinsic rewards.
> > > >
> > > > This is “just” a hypothesis, and because we are not the only paper evaluating these methods on MiniGrid (e.g. see [Raileanu and Rocktäschel’s (ICLR 2020) paper on RIDE](https://iclr.cc/virtual_2020/poster_rkg-TJBFPB.html)), it seems like a better place for this discussion would be a survey of exploration methods rather than a paper about a specific and novel exploration method like AMIGO, both in terms of focus and in terms of abiding by space constraints.

---

> > > > > ### Comment · AnonReviewer5 · 2020-11-23
> > > > > **Thanks! One more clarification**
> > > > >
> > > > > Thanks for the detailed discussion, here. Apologies, I should have responded sooner, but on my first pass I misread a key bit and only now am I realizing the use of asking one more clarification regarding this.
> > > > >
> > > > > In this discussion of partial/full observability, one thing that concerns me is that, in going from full -> partial, two transformations are being done.
> > > > >
> > > > > 1. The image is made agent-centric. As you discuss, this can lead to drastic differences in how certain intrinsic motivation signals behave.
> > > > > 2. The image is cropped to be a small (agent-centric) window. As you discuss, this can reduce the dimensionality and make some aspects of learning easier. On the other hand, it also drastically reduces the amount of information available.
> > > > >
> > > > > Do these experiments you run tease apart these effects? My impression is that they do not. Now, of course when using an agent-centric view, one must either crop or pad, and padding has the challenge of making the input space larger still, but it would have the (potentially very important) benefit of making all information available to the agent.
> > > > >
> > > > > In short, my lingering concern is that these two choices are being made jointly in the experiments, with somewhat opaque tradeoffs (I am intrigued by your suggestion of why it might be useful for both the agent and intrinsic motivation to have the same view...this isn't something that would be immediately apparent to me), and that these choices might drive baseline performance. Now, it's interesting that many approaches are fragile to this, and it makes a lot of sense that goal-based techniques might be less so.
> > > > >
> > > > > I've updated my score. Due to these concerns, I've not updated it higher, but I am certainly open to changing it based on further comments by authors/reviewers.

---

> > > > > > ### Author Response · Authors · 2020-11-23
> > > > > > **One more clarification in response**
> > > > > >
> > > > > > We are sure you are engaging in the interest of ensuring the peer review process meaningfully supports research of the highest standards. That said, we are somewhat disappointed that your score indicates our work does not merit publication. We respectfully think this might be due to a misunderstanding which we seek to clarify here in the hope you will feel comfortable supporting the paper given all the improvements made thanks to your feedback.
> > > > > > ### On the agent-centric view
> > > > > > When we use a “partial view”, we indeed crop (rather than pad) a small view from the perspective of the agent, as specified in the original MiniGrid environments (see https://github.com/maximecb/gym-minigrid for some examples). Please note that **we did not choose this implementation of a partial view arbitrarily: it is what the authors of the MiniGrid environment recommend to use**. Furthermore, we chose the settings for training the baselines in order to give them the **best possible chance to perform well** and to make the comparison with AMIGo as fair as possible. We really want to underline this point here: **if anything is “unfair” about the way we evaluate baselines, it is unfair to AMIGo, and maximally generous to the baselines**, thereby should give you greater confidence in our findings. We expand on this below.
> > > > > >
> > > > > > We chose the partial-intrinsic-reward-partial-policy setting because most of the baselines were designed and evaluated for this setting (with RIDE in this setting being the previous state of the art on MiniGrid). We also used the non-centered full view for the policy because this is what AMIGo uses so we wanted to provide the baselines with the same information in exactly the same format. In addition, we paired the non-centered full view for the policy with both partial and (non-centered) full view for the intrinsic reward in an attempt to compare with the strongest possible baselines. **In fact, training the baselines with agent-centric full views would not be a completely fair comparison since AMIGo uses a non-centered full view, and therefore has less access to information about where the agent is.** So we believe the setting we used provides a more fair comparison, by being less fair to our own method and more generous with the baselines.
> > > > > >
> > > > > > In addition, we are doubtful that the setting you propose (i.e. using an agent-centric full view) would lead to strictly better baseline performance. As we explained in our previous response, we do not believe that using agent-centric full views would work better for computing the intrinsic reward than using agent-centric partial views. Given that the intrinsic reward must be based on partial views, it could be problematic for the policy to be based on full views, as we also touched on in the previous response. To expand on this, imagine that the agent can be in very different states (i.e. full views) but still receive very similar intrinsic rewards if the partial views of the transition are the same (which is often the case in these environments). This means that the agent cannot take full advantage of the full view of the environment since the intrinsic reward does not depend on the full view. In particular, in very sparse reward settings such as our environments, this training regime is unlikely to lead to much better results than a partial-intrinsic-reward-partial-policy setting.
> > > > > >
> > > > > > ### Summary
> > > > > > Mindful of the 24h to complete this discussion, we have readily tried to respond to your concern above.
> > > > > >
> > > > > > We consider this paper successfully explains a new interesting method, thoroughly comparing against related work in an established setting, and demonstrably establishes a new state of the art. We think you have acknowledged this and while we agree there is always room for more extensive analysis, consider that we have put significant effort towards extensively studying the baselines to make a fair comparison for a 8 page conference paper. We are confident from our discussion we have made the case to convince you that "5" is too low a grade, and that you might be willing to change your mind and decide to strongly support our publication (noting that the scale of 1-10 leaves room for such assessments, e.g. 7 or 8, which both support the paper but state that it is not perfect). We hope you will agree, and again thank you for engaging in the discussion process.

---

> > > > > > > ### Comment · AnonReviewer5 · 2020-11-24
> > > > > > > **thanks! one more clarification**
> > > > > > >
> > > > > > > Thank you for your careful response on this, and I really appreciate your patience in sorting this out!
> > > > > > >
> > > > > > > If I could come around to be convinced that **"if anything is “unfair” about the way we evaluate baselines, it is unfair to AMIGo, and maximally generous to the baselines,"** then certainly I should upgrade my score! This is the crux of my concern; I'm worried that the particular choices made make the results a bit hard to interpret. A partially observed version of an RL environment can be much harder than its fully observed counterpart. Hence, it can be difficult to compare scores of a proposed algorithm that uses full observations with one that uses partial observations. Of course, you're simply giving the baselines an extra opportunity to succeed when making this comparison. However, these extra opportunities come with simultaneously manipulating agent-centric/not (some good arguments that intrinsic motivation in some cases perform very poorly when not agent-centric) and partial observability/full observability (partial observability adds major difficulties).
> > > > > > >
> > > > > > > So, as a result, I don't know what to make of these extra opportunities. What is clear is that in the non-centered full view, all of the baselines fall completely flat, whereas AMIGo achieves some success. This is great! Though our discussion has led me to wonder whether these pre-processing choices matter a great deal for the baselines, and these were simply the wrong ones. What I think is important is that we know we're not in a situation where the only way we were able to get the baselines to work at all was to make choices that considerably hurt performance, whereas other choices would give a different picture. If you would clarify two points you make above, that would be very helpful for me in order to have confidence in your statement that I bolded above.
> > > > > > >
> > > > > > > 1) "In fact, training the baselines with agent-centric full views would not be a completely fair comparison since AMIGo uses a non-centered full view, and therefore has less access to information about where the agent is." Could you say more about this? It's certainly the case that in order to center the view, you need to know where the agent is, so a preprocessing step would need to have this information provided. But this seems like a small thing to ask for, and if not having this is the *main* thing that kills baselines, that would be useful to know.
> > > > > > >
> > > > > > > 2) "In addition, we are doubtful that the setting you propose (i.e. using an agent-centric full view) would lead to strictly better baseline performance. **As we explained in our previous response, we do not believe that using agent-centric full views would work better for computing the intrinsic reward than using agent-centric partial views.** Given that the intrinsic reward must be based on partial views, it could be problematic for the policy to be based on full views, as we also touched on in the previous response." I'm a bit worried that this is very speculative. In particular, could you say more about the bolded part? I may be misunderstanding things, but in your previous response, agent-centric and partial observability seemed to be considered together. If agent-centric full views work about as well as agent-centric partial views for intrinsic motivation, then how would your quoted argument work?
> > > > > > >
> > > > > > > Thanks for bearing with me on this. I think that sorting out these particulars is really critical for interpreting the experiments and hence the whole work. I should say that I don't expect you to suddenly run more experiments! You've clearly done a lot of careful work -- I think that I would also be inclined to raise my score further if I were confident that these sorts of interpretation issues were to be discussed carefully in the paper.

---

> > > > > > > > ### Author Response · Authors · 2020-11-24
> > > > > > > > **Some closing replies and remarks (part 1)**
> > > > > > > >
> > > > > > > > Thank you for your further comments that help us improve our paper by avoiding potential sources of misunderstanding. We understand that you **“don't know what to make of these extra opportunities”**. We aim to elucidate this here, but before doing so, we wish to emphatically state: it would have been a fair comparison to just compare AMIGo against related work in six different difficult exploration settings. However, to ensure that we always benchmarked AMIGo in the harshest reasonably achievable manner, we always did a 4-way comparison of each baseline (i.e. in each observation setting) against AMIGo (in one observation setting). **This is a more rigorous form of comparison than what much of the related work in the literature does**, as it gives the most opportunities for success to baselines and gives the fewest (just one observation setting per environment) to AMIGo. We are adamant that **this is a rigorous assessment, which gives an undeniable empirical result**, and while there is always scope for further and deeper analysis, our experiments show a significant advantage in our method over related work, which in our view, in itself should be sufficient for publication.
> > > > > > > >
> > > > > > > > ### On a fair comparison with AMIGo
> > > > > > > > You write **”A partially observed version of an RL environment can be much harder than its fully observed counterpart. Hence, it can be difficult to compare scores of a proposed algorithm that uses full observations with one that uses partial observations.**
> > > > > > > > It may be true that partially observed versions of RL environments are harder to solve than fully observed versions, which is why we ran all baselines both in the partially observed setting and the same fully-observed setting as AMIGo, both for the policy and for the intrinsic motivation system. It is true that comparisons across observation settings is difficult, which is _precisely_ why we always did a 4-way comparison with AMIGo and took the best baseline results.
> > > > > > > >
> > > > > > > > Furthermore, you ask what we mean by “training the baselines with agent-centric full views would not be a completely fair comparison since AMIGo uses a non-centered full view, and therefore has less access to information about where the agent is”. By this, we mean that in the fully observed setting, the policy must not only learn what to do and how to do it, but also learn what it is controlling (i.e. what is the agent’s avatar in the environment, rather than another part of the environment it has no control over). In contrast, in agent-centric crops the information of the agent’s location is always implicitly provided, which brings us to your next point...
> > > > > > > >
> > > > > > > > You write **”It's certainly the case that in order to center the view, you need to know where the agent is, so a preprocessing step would need to have this information provided. But this seems like a small thing to ask for, and if not having this is the main thing that kills baselines, that would be useful to know.”** You are correct that we could provide this information about the agent’s location via an additional observation channel (i.e. an indicator vector the size of the map). The crucial points are:
> > > > > > > > 1. This is a simplifying assumption which we did not make for either AMIGo or the baselines, so the comparison stays fair.
> > > > > > > > 2. It is not standard practice in the related literature in MiniGrid to add such simplifying assumptions, and we aimed to replicate the same sort of experimental protocol used in related work in our own experiments.

---

> > > > > > > > > ### Comment · AnonReviewer5 · 2020-11-24
> > > > > > > > > **Thanks!**
> > > > > > > > >
> > > > > > > > > Thank you for your detailed responses. I think I'm satisfied now, and I've updated my score to reflect that.

---

> > > > > > > > ### Author Response · Authors · 2020-11-24
> > > > > > > > **Some closing replies and remarks (part 2)**
> > > > > > > >
> > > > > > > > (following [part 1 of our closing replies and remarks](https://openreview.net/forum?id=ETBc_MIMgoX&noteId=8d337YwqpEx)...)
> > > > > > > >
> > > > > > > > ### On the intrinsic reward
> > > > > > > > You write **“I may be misunderstanding things, but in your previous response, agent-centric and partial observability seemed to be considered together. If agent-centric full views work about as well as agent-centric partial views for intrinsic motivation, then how would your quoted argument work?”** We think there is a misunderstanding here about the use of these terms and we hope the following will clarify. What we are saying is that **the agent-centric full view will NOT work as well as agent-centric partial view (i.e. the one we refer to as partial observation in previous responses and throughout the paper) for similar reasons for which non-centered full view (i.e. the one we refer to as full observation throughout our paper) does not work as well as the agent-centric partial view**. As we explained in [our second response to your comments](https://openreview.net/forum?id=ETBc_MIMgoX&noteId=Kc-VxtHOUm6), if you use a full view, whether it is agent-centric or not, most of your inputs will not be affected by the agent’s actions so your intrinsic rewards will be largely uniform and won’t distinguish new, interesting or surprising states from others. Thus, it will not provide a good learning signal to the agent and training will reduce to having no intrinsic rewards. Please refer to [our second response](https://openreview.net/forum?id=ETBc_MIMgoX&noteId=Kc-VxtHOUm6) for more details about this. **Our argument was written for full view in comparison with agent-centric partial view, and thus applies to both agent-centric and non agent-centric full views**.
> > > > > > > >
> > > > > > > > We agree with you that it would indeed be helpful for readers to include a more in-depth discussion about these choices and their effect on learning. We will add a discussion about the two disentangled factors of variation of using partial / full views and agent-centric / non agent-centric views. However, we strongly believe that further analysis is significantly outside the scope of our paper and does not affect our results.
> > > > > > > >
> > > > > > > > ### Summary
> > > > > > > > To hopefully bring this discussion to a close, we thank you again for helping us significantly improve the paper’s clarity, and our own confidence in the findings. We summarize our key points here:
> > > > > > > > 1. Our paper forms **a new state of the art** relative to the related work compared against, on not one but several difficult (and previously unsolved) exploration tasks.
> > > > > > > > 2. Our experimental method makes **the same simplifying assumptions (or rather lack of them) as cited related work** which we directly compared against.
> > > > > > > > 3. We evaluated baselines we compare against in **a maximally generous setting for the baselines**, taking the maximum amongst all observation modalities per baseline, per task. We did no such thing with AMIGo, ensuring that it is benchmarked in the harshest and **most critical way possible**. We _still_ find it is the most competitive method in terms of sample complexity on medium environments (with comparable asymptotic performance) and in terms of overall performance on the harder environments (which the related work _cannot solve_).
> > > > > > > > 4. We are cognizant that there is always more variation that can be introduced in our experimental setting, yielding further experiments, but **the number of experiments and the rigour with which the comparison was made goes significantly beyond what was done in related work**, which we invite the reviewer to consult if they doubt our word.
> > > > > > > >
> > > > > > > > We therefore believe we have empirically, and unambiguously made the case for our method, and that the paper is worth publishing at this conference. We have made careful note throughout our discussion of some of the points you have brought up, which we assure you will meaningfully and positively influence follow up work will do on this method, but within the scope of this paper we hope you will agree we have said and done all that needs to be said in done. We now would dearly welcome your support for our paper, in which you have invested a significant amount of time helping us improve presentation and clarity, and thank you again for your efforts and kind attention to our work during this discussion.

---

> > > > ### Author Response · Authors · 2020-11-17
> > > > **Further response to your comments and questions (part 2)**
> > > >
> > > > (following [part 1 of our further response to your comments](https://openreview.net/forum?id=ETBc_MIMgoX&noteId=Kc-VxtHOUm6))
> > > >
> > > > ### On using partial or full observations for the policy
> > > > Regarding, now, the question of why some baselines work better using partial observations for the policy, we believe the high-dimensional input could make it harder to learn, under some approaches, in the fully observed setting. As we explained above, the intrinsic reward can benefit from using partial observations. If the intrinsic reward uses a partial view while the policy uses a full view, this might create some inconsistencies, because agents are rewarded on the effect their actions have on the agent-centric view, but their actions have a different effect on the fully observed view which determines how they interact with the environment globally. If both the intrinsic reward and the policy are based on a partial view, the agent is encouraged to learn local rules that are generally good no matter the full state of the env (which is typically enough for these environments which were originally created for learning from partial observations).
> > > >
> > > > As with the discussion above, we believe this analysis is best left for a survey paper given the space constraints here, but we hope this was useful to help the reviewer to interpret the results.
> > > >
> > > > ### On the need for additional experiments
> > > > To be clear, Tables 2-5 already disentangle the effect using full vs partial observability for the agent and the intrinsic motivation, respectively. This serves as an ablation study against the effect of, amongst other things, taking agent-centric crops, so these factors are already studied in isolation. As a result, we are unsure what further experiments you are calling for, since you seem to be asking for these factors of variation to be studied in isolation, which is precisely what is being done in our experiments (with the max of the scores per environment per baseline being reported in Table 1 as a means of comparing AMIGo against each baseline in their best performing observational setting). If we have misunderstood you, please let us know and we will gladly attempt any experiments you ask for, time and resources permitting.
> > > >
> > > > ### Summary
> > > > Thank you again for your insightful questions and comments. We hope our further response has clarified any outstanding issues. As discussed above, we are unsure the matters discussed here are essential to include in this paper (rather than, say, a survey of intrinsic motivation methods), but we aimed to at least ensure your interpretation and ours are aligned. Regarding further experiments, we hope to have clearly explained why we think the ablation study implicitly present in Tables 2-5 caters to your question here. We look forward to hearing from you if you have further comments, and naturally hope for your strong support for the paper, which we are confident has been further improved thanks to your feedback.

---

### Author Response · Authors · 2020-11-18
**Paper updated**

Thank you, everyone, for your helpful comments. We have made the following updates to the paper on the basis of your feedback, and uploaded the revised copy to OpenReview.

* Added references and discussion in Section 2 wrt intrinsic motivation
* Added references and discussion in Section 2 wrt Curriculum learning
* Clarifications regarding both the experimental setting and the notion of goals used in section 3.3.
* We ran all experiments projected to return 0.0 mean expected reward and confirmed, empirically, that the projects were correct. We have updated Tables 2-5 of Appendix A to reflect that all results are exact.
* We added clarification about the temporal frequency with regard to temporal frequency with which the teacher is trained in Section 3.2.
* We added details about initial experiments with fixed thresholds in Section 3.2, and about the need to vary the threshold across environments (as one might expect).
* We added a further illustrative example of the sort of goal incentives episode boundary awareness provides the teacher policy in Section 3.4.

Please note that there are certainly further stylistic improvements, and additional detail, that can be incorporated into the main body of the paper. We have reserved half a page in case additional crucial changes emerge during the discussion, and if none arise we will use that space for those additional details. For the time being, we have focussed on the main changes the reviewers asked to ensure that crucial ambiguities are resolved.

---

### Decision · Program_Chairs · 2021-01-07
**Final Decision**

**Decision:**

Accept (Poster)

**Comment:**

This paper was reviewed by four experts in the field. Based on the reviewers' feedback, the decision is to recommend the paper for acceptance to ICLR 2021. The reviewers did raise some valuable concerns that should be addressed in the final camera-ready version of the paper. The authors are encouraged to make the necessary changes and include the missing references.